# Functional metagenomics-guided discovery of potent Cas9 inhibitors in the human microbiome

**Kevin J Forsberg[1]\*, Ishan V Bhatt[1], Danica T Schmidtke[1], Kamyab Javanmardi[2], Kaylee E Dillard[2], Barry L Stoddard[1], Ilya J Finkelstein[2,3], Brett K Kaiser[1,4], Harmit S Malik[1,5]**

[1]Division of Basic Sciences, Fred Hutchinson Cancer Research Center, Seattle, United States; [2]Department of Molecular Biosciences and Institute of Cellular and Molecular Biology, University of Texas at Austin, Austin, United States; [3]Center for Systems Biology and Synthetic Biology, University of Texas at Austin, Austin, United States; [4]Department of Biology, Seattle University, Seattle, United States; [5]Howard Hughes Medical Institute, Fred Hutchinson Cancer Research Center, Seattle, United States

**Abstract** CRISPR-Cas systems protect bacteria and archaea from phages and other mobile genetic elements, which use small anti-CRISPR (Acr) proteins to overcome CRISPR-Cas immunity. Because Acrs are challenging to identify, their natural diversity and impact on microbial ecosystems are underappreciated. To overcome this discovery bottleneck, we developed a high-throughput functional selection to isolate ten DNA fragments from human oral and fecal metagenomes that inhibit *Streptococcus pyogenes* Cas9 (SpyCas9) in *Escherichia coli*. The most potent Acr from this set, AcrIIA11, was recovered from a *Lachnospiraceae* phage. We found that AcrIIA11 inhibits SpyCas9 in bacteria and in human cells. AcrIIA11 homologs are distributed across diverse bacteria; many distantly-related homologs inhibit both SpyCas9 and a divergent Cas9 from *Treponema denticola*. We find that AcrIIA11 antagonizes SpyCas9 using a different mechanism than other previously characterized Type II-A Acrs. Our study highlights the power of functional selection to uncover widespread Cas9 inhibitors within diverse microbiomes.
DOI: https://doi.org/10.7554/eLife.46540.001

**\*For correspondence:**
kforsber@fredhutch.org

**Competing interests:** The authors declare that no competing interests exist.

## Introduction

CRISPR-Cas adaptive immune systems are present in many bacterial and archaeal genomes (*Makarova et al., 2015*; *Burstein et al., 2016*), where they protect their hosts from infection by phages (*Barrangou et al., 2007*), plasmids (*Marraffini and Sontheimer, 2008*), and other mobile genetic elements (MGEs) (*Zhang et al., 2013*). CRISPR-Cas systems mediate this defense by incorporating short (~30 bp) spacer sequences from invading genomes into an immunity locus in the host genome. These spacer sequences are then expressed and processed into CRISPR RNAs (crRNAs) that, together with various Cas nucleases, mediate homology-dependent restriction of invading genomes. CRISPR-Cas systems are classified into six types (I, II, III, etc.) and 30 subtypes (I-A, I-B, II-C, etc.) on the basis of functional differences, phylogenetic relatedness, and locus organization (*Makarova et al., 2018*).

In response to restriction, phages and other MGEs have evolved dedicated CRISPR-Cas antagonists, called anti-CRISPRs (Acrs) (*Bondy-Denomy et al., 2013*), which can promote phage infection, enable horizontal gene transfer (HGT), and thus shape microbial ecosystems (*Borges et al., 2017*; *Pawluk et al., 2017a*). Acrs that inhibit the type II Cas9 (*Pawluk et al., 2016a*; *Rauch et al., 2017*)

**eLife digest** Viruses that attack bacteria are known as bacteriophages, or phages for short. Bacteria have developed an antiviral immune system called CRISPR-Cas that works by targeting particular genetic sequences, such as those of an invading phage, for destruction. To counteract this immune system, phages have evolved proteins that can block CRISPR-Cas known as anti-CRISPRs.

Researchers have studied the CRISPR-Cas bacterial defense systems intensively over the past decade but much less is known about anti-CRISPRs. For example, the natural diversity and prevalence of anti-CRISPRs is still unknown, and identifying these proteins has proven difficult.

To address this gap, Forsberg et al. developed a technique to identify new anti-CRISPRs based on their ability to inhibit CRISPR-Cas activity. The method relies on three elements. First, a piece of DNA that lets bacteria resist a specific antibiotic. Second, a test piece of DNA that might code for an anti-CRISPR. Third, a CRISPR-Cas system designed to target and destroy the antibiotic resistance DNA. The three elements are put into bacteria, and two things can happen. If the 'test DNA' does not code for an anti-CRISPR, then the CRISPR-Cas system destroys the antibiotic resistance DNA and the bacteria die when exposed to the antibiotic. On the other hand, if the test DNA does code for an anti-CRISPR, it will inhibit the CRISPR-Cas system and the antibiotic resistance DNA will remain intact. This means that the bacteria will survive when grown in the antibiotic, and new anti-CRISPRs can be found by examining the test DNA in those bacteria.

Forsberg et al. employed this strategy to screen a huge library of DNA pieces, uncovering several new anti-CRISPRs. They then focused on an anti-CRISPR that was very common in the human gut called AcrIIA11. Biochemical characterization showed that AcrIIA11 inhibited CRISPR-Cas via a different mechanism from other known anti-CRISPRs. Moreover, it could inhibit CRISPR-Cas systems from many different bacteria.

The potential to systematically identify anti-CRISPRs able to resist any bacterium's CRISPR-Cas defense system could lead to the design of phages that can infect bacteria which are otherwise difficult to destroy. In the future, these phages could be used to clear antibiotic-resistant infections. Beyond its role as an antiviral system in bacteria, CRISPR-Cas is a widely used tool for genetic modification in biomedical research. Using anti-CRISPRs to regulate where, how, and when CRISPR-Cas systems act could make their many emerging applications safer and more effective.
DOI: https://doi.org/10.7554/eLife.46540.002

and type V Cas12 systems (*Marino et al., 2018*; *Watters et al., 2018*) have also garnered significant interest in biotechnology, with demonstrated utility for reducing off-target Cas9 lesions (*Shin et al., 2017*), suppressing gene drives (*Basgall et al., 2018*), and precisely controlling synthetic gene circuits (*Nakamura et al., 2019*).

Work over the past decade has revealed much about the activity (*Hille et al., 2018*), distribution (*Makarova et al., 2015*), and evolution (*Koonin and Makarova, 2017*) of CRISPR-Cas systems. In contrast, comparatively little is known about Acr diversity and function. Known Acrs inhibit only seven of the 30 CRISPR-Cas subtypes (*Bondy-Denomy et al., 2018*; *Makarova et al., 2018*), though it is quite likely that unidentified Acrs antagonize the remaining 23 groups (*Pawluk et al., 2017a*). In cases where antagonists of CRISPR-Cas systems have been identified, it is also likely that many additional, undiscovered Acrs exist to antagonize those systems (*Watters et al., 2018*). These undiscovered Acrs likely act via a diversity of mechanisms to influence gene flow and phage dynamics in microbial communities (*van Belkum et al., 2015*; *Westra et al., 2016*; *Borges et al., 2017*), and may unlock new modes of manipulating phage- and CRISPR-Cas-enabled technologies (*Sheth et al., 2016*; *Knott and Doudna, 2018*).

Previous efforts to discover *acr*s have relied on phage genetics (*Bondy-Denomy et al., 2013*; *Pawluk et al., 2014*; *Hynes et al., 2017*; *He et al., 2018*), linkage to conserved genes (*Pawluk et al., 2016a*; *Pawluk et al., 2016b*; *Hynes et al., 2018*; *Lee et al., 2018*; *Marino et al., 2018*) or the presence of a self-targeting CRISPR spacer, which would create an unstable autoimmune state if not for the presence of an Acr protein (*Rauch et al., 2017*; *Marino et al., 2018*; *Watters et al., 2018*). These clever strategies have revealed many candidate *acr*s. However, they have also highlighted the difficulty of finding new *acr* genes based on homology, since *acr*s share

little sequence conservation (*Sontheimer and Davidson, 2017*). As a result, most *acr*s almost certainly lie unrecognized among the many genes of unknown function in phages, plasmids, and other MGEs (*Hatfull, 2015*).

To overcome the challenges associated with anti-CRISPR discovery, we devised a functional metagenomic selection that identifies *acr* genes from any cloned DNA, based on their ability to protect a plasmid from CRISPR-Cas-mediated destruction. Recently, *Uribe et al. (2019)* independently developed a similar Acr search strategy. Because functional metagenomics selects for a function of interest from large clone libraries (*Handelsman, 2004*), it is well-suited to identify individual genes like *acr*s that have strong fitness impacts (*Iqbal et al., 2014*; *Forsberg et al., 2015*; *Forsberg et al., 2016*; *Genee et al., 2016*). This approach may be particularly useful for Acr discovery because Acrs are expressed from single genes and function readily in many genetic backgrounds (*Pawluk et al., 2016a*; *Rauch et al., 2017*).

Using this functional selection, we find that many unrelated metagenomic clones from human oral and gut microbiomes protect against *Streptococcus pyogenes* Cas9 (SpyCas9), the variant used most commonly for gene editing applications (*Knott and Doudna, 2018*). We identify a broadly distributed but previously undescribed Acr from the most potent SpyCas9-antagonizing clone in our libraries. This Acr, named AcrIIA11, binds both SpyCas9 and double-stranded DNA (dsDNA) and exhibits a novel mode of SpyCas9 antagonism, protecting both plasmids and phages from immune restriction.

## Results

### A functional metagenomic selection for type II-A anti-CRISPRs

We designed a functional selection that can isolate rare *acr*s from complex metagenomic libraries. We based this selection on the ability of an *acr* gene product to protect a plasmid, which bears an antibiotic resistance gene, from being cleaved by SpyCas9 (*Figure 1A*). By screening metagenomes, our selection interrogates core bacterial genomes as well as DNA from the phages, plasmids, and other mobile genetic elements that infect these bacteria, which must contend with CRISPR-Cas immunity. Because most DNA inserts in large metagenomic libraries lack an *acr*, most clones in a library should be susceptible to SpyCas9-mediated destruction. However, those few DNA inserts that encode and express a functional Acr will resist SpyCas9 and can be recovered using the antibiotic resistance conferred by the plasmid they protect.

In our *acr* selection scheme (*Figure 1—figure supplement 1*), we targeted an inducible SpyCas9 nuclease to a kanamycin resistance ($Kan^R$) gene on a plasmid used to construct the metagenomic libraries. Anticipating that resistance to SpyCas9 cleavage could arise readily via point mutations in target sites, we targeted SpyCas9 to two distinct loci within the $Kan^R$ gene (*Figure 1B*) to reduce the frequency of these 'escape' plasmids. We transformed metagenomic libraries into an *Escherichia coli* strain that contained SpyCas9 under an arabinose-inducible promoter. After the cells were allowed to recover, we grew them overnight with arabinose to induce SpyCas9 expression. During this phase, cells were not exposed to kanamycin and thus were not under selection to maintain the metagenomic library bearing the $Kan^R$ gene. We then plated cells on solid media with kanamycin, killing cells in which SpyCas9 cleaved the $Kan^R$ gene and allowing the recovery of metagenomic DNA from the few remaining Kan^R colonies.

Our analysis revealed that SpyCas9 loss-of-function mutants (*Figure 1C*) dominated the Kan^R population in early, single-iteration experiments, occurring in approximately $10^{-4}$ to $10^{-5}$ transformants (*Figure 1—figure supplement 2*, *Supplementary file 1* table S1). We therefore added a second iteration of SpyCas9 selection, reasoning that the loss-of-function rate would remain constant across iterations, whereas *acr*-encoding clones would be enriched by a factor of ~$10^4$ with each iteration. In theory, this enables us to select for an *acr* gene even if it is present just once in a library of ~$10^7$ clones. To accomplish these two rounds of SpyCas9 selection, we purified total plasmid DNA from Kan^R colonies following one iteration and removed the original SpyCas9 expression plasmid via digestion with I-SceI, a homing endonuclease that cleaves an 18 bp target sequence. We then transformed the surviving metagenomic plasmids into a fresh SpyCas9-expressing strain and exposed them to SpyCas9 selection a second time (*Figure 1—figure supplement 1*). As expected,

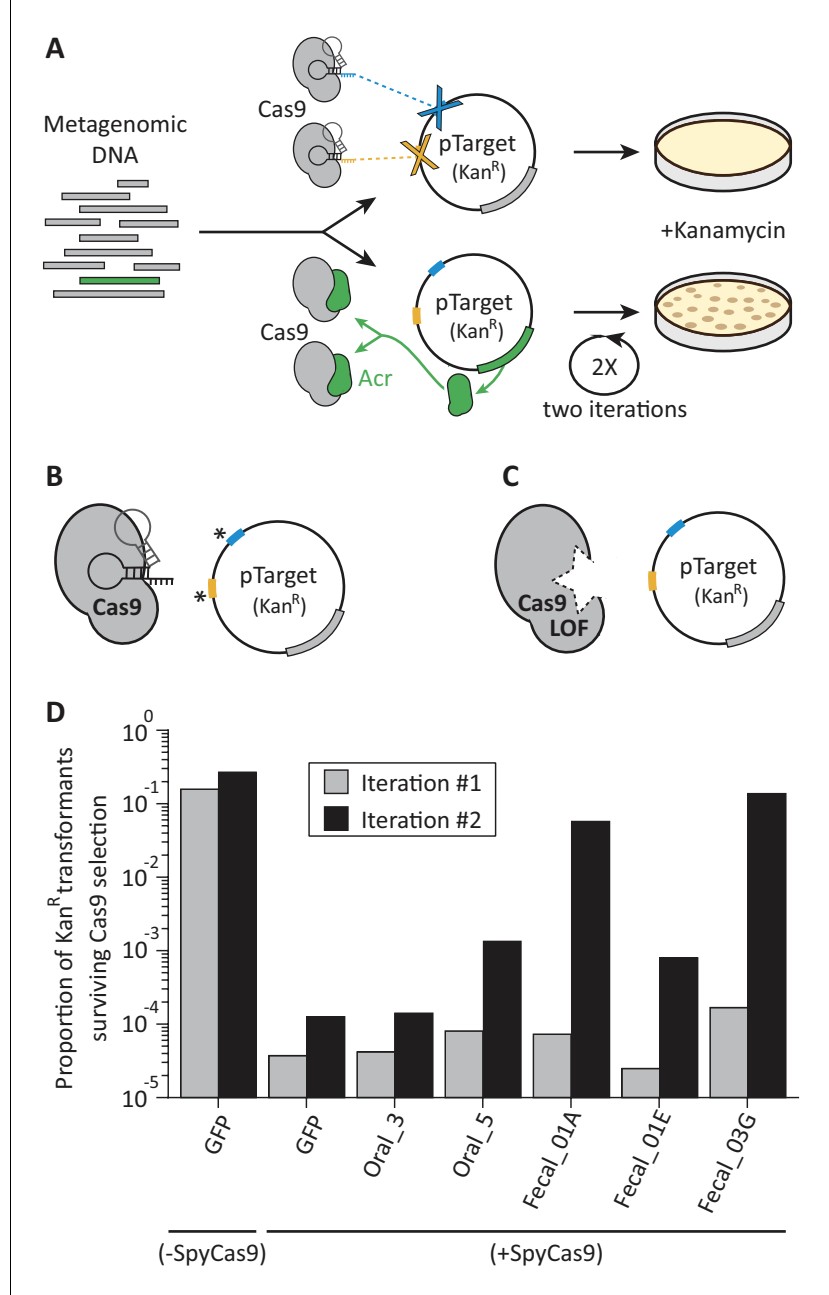

**Figure 1.** A functional metagenomic selection for type II-A anti-CRISPRs. (**A**) In a metagenomic library, plasmids without *acr*s are targeted by SpyCas9 and eliminated. Those plasmids that carry *acr*s (green) withstand SpyCas9 and can be recovered via kanamycin selection. However, individual target site mutations, indicated by an asterisk (**B**) or SpyCas9 loss of function (LOF) mutations (**C**) allow plasmids to evade SpyCas9 independent of their metagenomic DNA insert. To reduce these major sources of false positives, we employ two plasmid target sites (blue, yellow) and two rounds of selection. (**D**) Relative to a GFP control, some metagenomic libraries protect plasmids after two iterations rounds of SpyCas9 selection. Each bar represents a single experiment.

DOI: https://doi.org/10.7554/eLife.46540.003

The following figure supplements are available for figure 1:

**Figure supplement 1.** A functional selection for type II-A anti-CRISPRs.

DOI: https://doi.org/10.7554/eLife.46540.004

**Figure supplement 2.** SpyCas9 loss-of-function mutations are the major source of false-positives during *acr* selection.

DOI: https://doi.org/10.7554/eLife.46540.005

*Figure 1 continued on next page*

*Figure 1 continued*

**Figure supplement 3.** Coverage of assembled contigs by library.
DOI: https://doi.org/10.7554/eLife.46540.006

adding a second iteration of SpyCas9 selection resulted in a significant enrichment for *bona fide* SpyCas9 antagonists above background (*Figure 1D*).

## SpyCas9 antagonism in human oral and fecal metagenomes

We used our functional selection to search for *acr*s in five metagenomic libraries: two oral metagenomes from Yanomami Amerindians (*Clemente et al., 2015*) and three fecal DNA metagenomes from peri-urban residents of Lima, Peru (*Pehrsson et al., 2016*). We subjected each of these libraries, with an estimated $1.3 \times 10^6$ - $3.4 \times 10^6$ unique clones per library, to SpyCas9 selection (*Supplementary file 1* table S2). For each library, we observed a $10^4$ to $10^5$-fold reduction in the proportion of Kan^R colony forming units (CFU) following one iteration of SpyCas9 selection. This value matches the reduction in Kan^R CFU seen for a GFP control (*Figure 1D*) and the empirically determined frequency of SpyCas9 loss-of-function mutations (*Figure 1—figure supplement 2*, *Supplementary file 1* table S1). Since Kan^R is a measure of plasmid retention, this result indicates that most clones in each library cannot inhibit SpyCas9. Following a second round of SpyCas9 selection for each library, metagenomic inserts were amplified from pooled Kan^R colonies by PCR, deep-sequenced, and assembled de novo using PARFuMS (Parallel Assembly and Re-annotation of Functional Metagenomic Selections) (*Forsberg et al., 2012*). We used read-coverage over each assembled contig to estimate its abundance following selection. After quality-filtering (*e.g.* removal of low-abundance contigs), we recovered a total of 51 contigs across all five libraries that putatively antagonize SpyCas9 (*Figure 1—figure supplement 3*, *Supplementary file 1* tables S3, S4).

After two rounds of SpyCas9 selection, two libraries (Oral_5, Fecal_01A) seemed more likely to contain new *acr*s than the other three (Oral_3, Fecal_01E, Fecal_03G). The Oral_3 library poorly withstood SpyCas9 targeting, so was not studied further (*Figure 1D*). Although the Fecal_03G library completely resisted SpyCas9 (*Figure 1D*), subsequent analysis revealed that this was almost entirely due to a single clone that acquired mutations in both Cas9 target sites (*Figure 1B*, *Figure 1—figure supplement 3*, *Supplementary file 1* table S3). Accordingly, just one contig from this library passed quality filters. In contrast, the Fecal_01E library showed intermediate SpyCas9 resistance and was largely devoid of 'escape' mutations (*Figure 1D*, *Supplementary file 1* table S3). However, only one of the 18 contigs from this library was found to be phage-associated (*Supplementary file 1* table S4). Because *acr*s are expected to originate in phages and MGEs (*Borges et al., 2017*; *Pawluk et al., 2017a*; *Sontheimer and Davidson, 2017*), we predicted that contigs from Fecal_01E were unlikely to contain *bona fide acr*s and did not prioritize them in this study. These contigs may nonetheless encode anti-SpyCas9 activity, perhaps via Cas9 regulatory factors employed by host bacteria rather than MGEs (*Høyland-Kroghsbo et al., 2017*; *Faure et al., 2019*), and therefore could represent a useful resource for probing host regulation of SpyCas9 activity.

Contigs from the Oral_5 and Fecal_01A libraries looked most promising for *acr* discovery. These libraries conferred SpyCas9 resistance at levels 10-fold to 1,000-fold above background (*Figure 1D*). Moreover, plasmids containing these contigs had few escape mutations in Cas9 target sites, so they likely withstood SpyCas9 due to functions encoded by their DNA inserts (*Supplementary file 1* table S3). Finally, many contigs from these libraries were found to be phage-associated (30%; *Figure 2A*, *Supplementary file 1* table S4) and many encoded genes of unknown function (*Figure 2—figure supplement 1*); these are both hallmarks of known *acr*s (*Borges et al., 2017*; *Pawluk et al., 2017a*; *Sontheimer and Davidson, 2017*). The contigs identified belonged to 19 bacterial genera. Intriguingly, Cas9 homologs are found in 15 of these 19 bacterial genera (79%, *Supplementary file 1* table S4), a significant enrichment over the 12% of bacterial genera that harbor Cas9 (567/4822) in a representative set of ~24,000 bacterial genomes (*Mendler et al., 2018*) (p=$8 \times 10^{-20}$, $\chi^2$ test). This enrichment strongly supports our hypothesis that many of the recovered contigs encode true *acr*s rather than artifacts of functional selection, since phages and MGEs must encounter Cas9 to benefit from the protective effect of *acr*s.

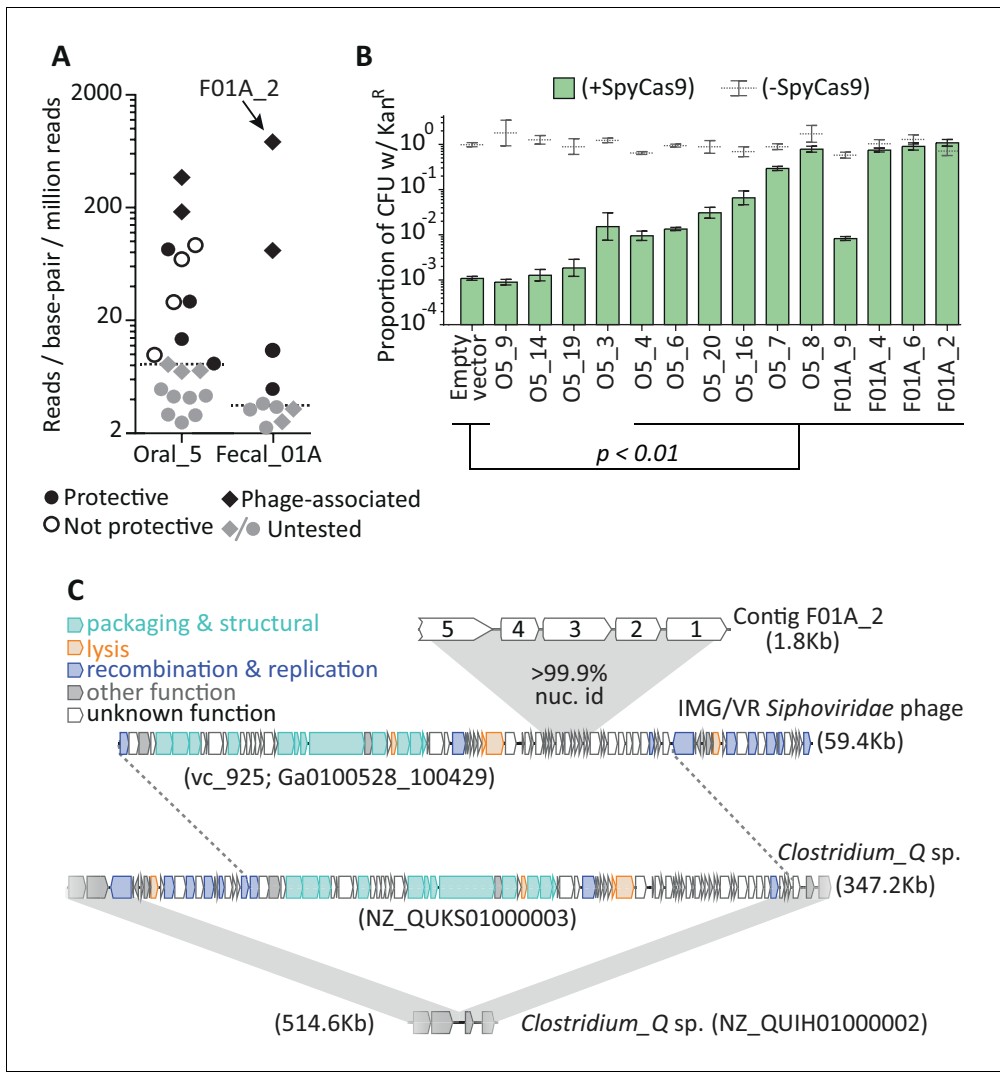

**Figure 2.** Metagenomic DNA inserts antagonize SpyCas9. (**A**) The read-coverage of all contigs from libraries Oral_5 and Fecal_01A is shown. Dashed lines depict median coverage for each library. Relative coverage can be used as a proxy for contig abundance within a library but cannot be used for meaningful comparisons across libraries. (**B**) Upon individual re-testing, six oral and four fecal inserts protect a plasmid from SpyCas9; p-values indicate comparisons to an empty vector in SpyCas9 inducing conditions and are adjusted for multiple hypotheses (Student's t-test, n = 3); + /- refers to SpyCas9 induction. Error bars depict standard error of the mean. (**C**) Contig F01A_2 is from a *Siphoviridae* phage which lysogenizes *Clostridium_Q* sp. Gray, shaded regions depict key regions of near perfect (>99%) nucleotide identity. Homology between the *Clostridium_Q* prophage and the IMG/VR phage is not depicted as these phages are nearly identical (except for one translocation or assembly-artefact involving much of the recombination and replication machinery). All accession numbers denote NCBI Genbank IDs except for the *Siphoviridae* phage, for which we use IMG/VR convention and indicate viral_cluster with the scaffold_id in parentheses. Contig lengths are depicted next to each sequence.

DOI: https://doi.org/10.7554/eLife.46540.007

The following figure supplements are available for figure 2:

**Figure supplement 1.** NCBI taxonomies and gene functions recovered from all SpyCas9-antagonizing contigs.

DOI: https://doi.org/10.7554/eLife.46540.008

**Figure supplement 2.** The genes on contig F01A_2 encode small accessory proteins of *Lachnospiraceae* phages.

DOI: https://doi.org/10.7554/eLife.46540.009

To confirm that the Oral_5 and Fecal_01A libraries encoded Acrs, we re-cloned the most abundant contigs from these libraries into new plasmid and strain backgrounds and tested them individually for SpyCas9 resistance. This step eliminated potential mutations to the plasmid backbone, the *spycas9* gene, or the host genome, which may have accounted for SpyCas9 resistance in the original screen. For 10 of 14 re-cloned contigs, the DNA insert still protected its parent plasmid from SpyCas9, confirming the power of our selection to identify novel *acr*s from metagenomic DNA (*Figure 2A and B*). Intriguingly, none of the contigs recovered by functional selection encoded homologs of previously identified *acr*s, nor did they contain homologs of any known *acr*-associated (*aca*) genes (*Bondy-Denomy et al., 2018*).

## Discovery of a widespread, broadly-acting Cas9 antagonist: AcrIIA11

Among the 10 contigs we confirmed as inhibiting SpyCas9, several features made us focus on a single contig, F01A_2 (*Figure 2A*). First, the F01A_2 contig was recovered from our iterative selection of the Fecal_01A library, which conferred near-complete protection against SpyCas9 (*Figure 1D*). Second, the F01A_2 contig was by far the most abundant contig from this library (with coverage 216-fold above the median coverage in the library), suggesting that it outperformed other contigs during selection. In our re-testing, we confirmed that F01A_2 completely inhibited SpyCas9 activity (*Figure 2B*). Finally, F01A_2 shares near-perfect nucleotide identity (>99.9%) with a *Siphoviridae* phage. This phage infects bacteria from the genus *Clostridium_Q*, where it is found as a prophage in the genomes of some strains but not in those of close relatives (*Figure 2C*), suggesting that it is an actively circulating phage. *Clostridium_Q* is a member of the family *Lachnospiraceae* described recently in a reassessment of bacterial taxonomy (*Parks et al., 2018*) and includes the species *Clostridium symbiosum*, an important biomarker associated with colorectal cancer progression (*Xie et al., 2017*; *Thomas et al., 2019*). As is typical for Acr-encoding loci, the five open reading frames (ORFs) on F01A_2 are small, map to accessory regions of phage genomes, and appear to routinely undergo HGT (*Figure 2—figure supplement 2*).

To identify the ORF(s) in F01A_2 responsible for SpyCas9 antagonism, we introduced an early stop codon into each of the five predicted ORFs in the contig and re-tested this set of null mutants for anti-Cas9 activity. *Orf_3* completely accounted for SpyCas9 inhibition: a null mutation in *orf_3* reduced the frequency of Kan$^R$$10^5$-fold, to the level of an empty-vector control (*Figure 3A*). This ORF, which we named *acrIIA11*, was sufficient for SpyCas9 antagonism, protecting a target plasmid (*Figure 3B*) from SpyCas9 approximately as well as *acrIIA4*, a potent inhibitor of SpyCas9 used in multiple gene-editing applications (*Rauch et al., 2017*; *Shin et al., 2017*; *Basgall et al., 2018*; *Nakamura et al., 2019*).

We also investigated the ability of *acrIIA11* to restore phage infection in the face of SpyCas9-mediated immunity. We tested SpyCas9's ability to prevent Mu phage infection in the absence of any Acr or in the presence of either *acrIIA4* or *acrIIA11*. When SpyCas9 is equipped with a non-targeting crRNA that does not recognize Mu's genome, the phage successfully infects *E. coli*. However, when SpyCas9 contains a Mu-targeting crRNA, its expression robustly inhibits phage infection, provided that no Acr is present. When either *acrIIA11* or *acrIIA4* is expressed, SpyCas9 immunity is nearly completely abolished, confirming the ability of these Acrs to inhibit SpyCas9 and restore phage infection (*Figure 3C*, *Figure 3—figure supplement 1*).

*AcrIIA11* bears no discernible homology to any previously identified *acr* (*Borges et al., 2017*; *Pawluk et al., 2017a*; *Lee et al., 2018*; *Uribe et al., 2019*). Since *acrIIA11* is a newly discovered *acr*, we wished to determine its distribution in nature. We therefore searched for homologs in NCBI and in IMG/VR, a curated database of cultured and uncultured DNA viruses (*Paez-Espino et al., 2017*). We identified many proteins homologous to AcrIIA11 in both phage and bacterial genomes and focused on a high-confidence set of homologs, those with ≥35% amino acid identity that cover ≥75% of AcrIIA11's 182 amino acid sequence (*Figure 4—figure supplement 1*). AcrIIA11 homologs have a wider phylogenetic distribution than most previously identified type II-A anti-CRISPRs (*Figure 4A*) and span multiple bacterial phyla (*Figure 4—figure supplement 2*, *Supplementary file 1* table S5). We made a phylogenetic tree of AcrIIA11 homologs and found that they clustered into three monophyletic groups (*Figure 4B*). AcrIIA11 homologs from each group were identified in a variety of MGEs (*Figure 4C*), consistent with our observation that AcrIIA11 routinely undergoes HGT (*Figure 2—figure supplement 2*, *Figure 4—figure supplement 3A*). Despite these signatures of HGT, the three groups on the AcrIIA11 phylogenetic tree (*Figure 4B*) correspond

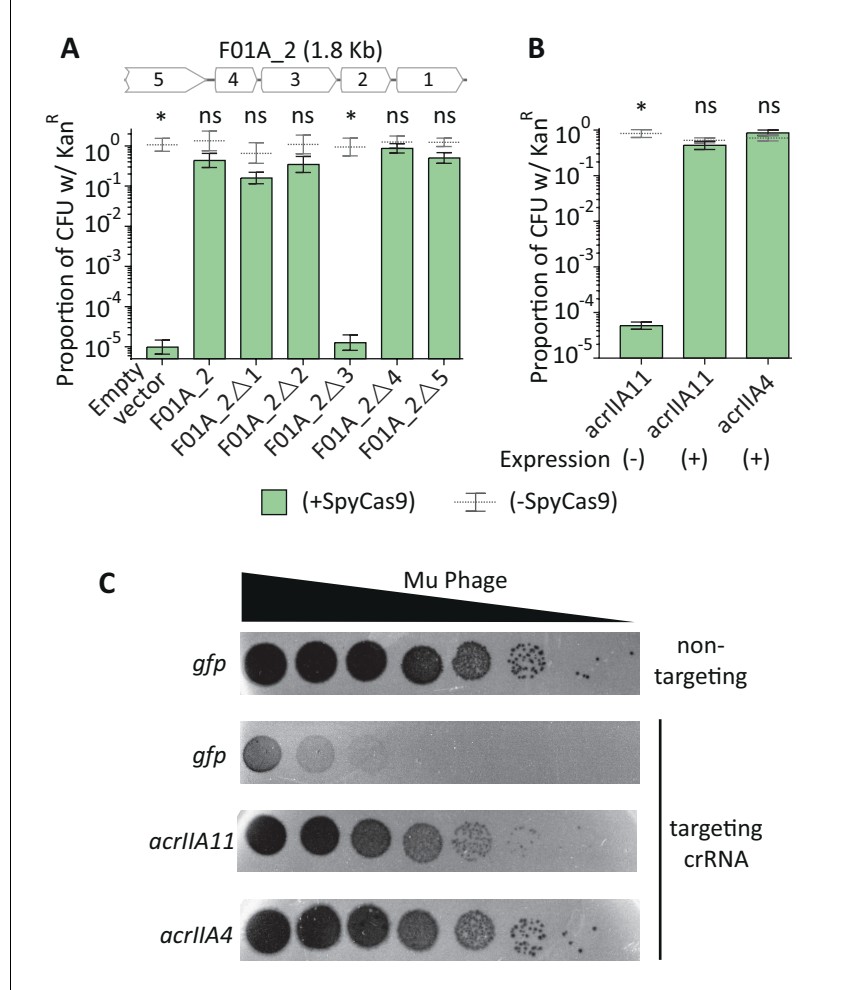

**Figure 3.** AcrIIA11 protects plasmid and phage from SpyCas9. (**A**) The F01A_2 contig is depicted above the bar chart. Delta symbols (Δ) indicate early stop codons in each gene of the contig. Only the third gene on contig F01A_2 is necessary for SpyCas9 antagonism. (**B**) Induction of the third gene, named *acrIIA11*, is sufficient for SpyCas9 antagonism, protecting a plasmid as well as *acrIIA4*. Asterisks in (**A**) and (**B**) depict statistically significant differences in plasmid retention between SpyCas9-inducing and non-inducing conditions (Student's t-test, $p < 0.01$, n = 3; p-values were corrected for multiple hypotheses and 'ns' indicates non-significance ($p > 0.05$). Error bars depict standard error of the mean. (**C**) Mu phage fitness, measured by plaquing on *E. coli* expressing Mu-targeting SpyCas9, is measured in the presence of *gfp*, *acrIIA11*, or *acrIIA4* via serial ten-fold dilutions (also see replicated in *Figure 3—figure supplement 1*). Based on a non-targeting (n.t.) crRNA control, we conclude that SpyCas9 confers ~$10^5$ fold protection against phage Mu in these conditions. Both *acrIIA11* and *acrIIA4* significantly enhance Mu fitness by inhibiting SpyCas9.

DOI: https://doi.org/10.7554/eLife.46540.010

The following figure supplement is available for figure 3:

**Figure supplement 1.** *AcrIIA11* protects phage from SpyCas9.
DOI: https://doi.org/10.7554/eLife.46540.011

directly to the three bacterial taxonomic clades in which AcrIIA11 is found (*Figure 4A*, *Figure 4—figure supplement 2*). This concordance between gene and species clusters indicates that, while HGT of *acrIIA11* may readily occur across short phylogenetic distances, intra-class and intra-phylum gene flow is rare, in contrast to what has been observed for some other *acr*s (*Uribe et al., 2019*).

AcrIIA11 homologs are only found in bacterial taxa that are highly diverged from *Streptococcus*. Nevertheless, AcrIIA11 potently inhibits SpyCas9, even though SpyCas9 is quite divergent from the type II-A Cas9 proteins found in AcrIIA11-encoding taxa. For instance, the only Cas9 protein in *Clostridium_Q* (CqCas9, NCBI accession CDD37961), AcrIIA11's genus-of-origin, shares just 32%

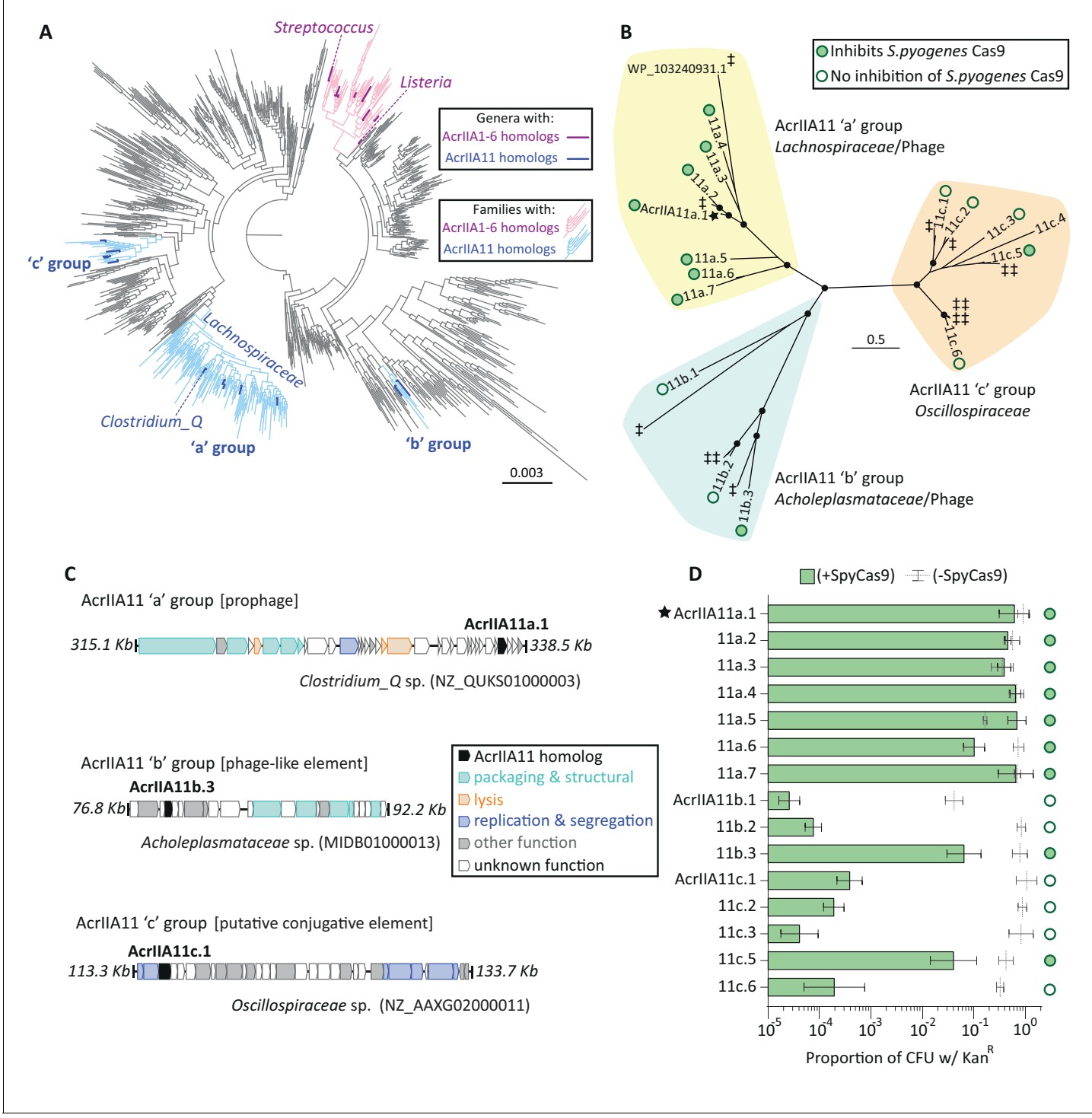

**Figure 4.** Diverse AcrIIA11 homologs inhibit SpyCas9. (A) A phylogenetic tree of Firmicutes and related phyla that shows the widespread dissemination of *acrIIA11* relative to other type II-A *acrs*. Each tip represents a bacterial genus; families colored light blue contain *acrIIA11* homologs while families colored pink contain homologs of *acrIIA1 - acrIIA6*. Genera are colored dark blue or magenta if *acr* homologs could confidently be assigned to this phylogenetic resolution. This tree also depicts the phylogenetic placement of the following genera: *Clostridium_Q*, *Streptococcus*, and *Listeria* (the genus in which the most type II-A *acrs* have been discovered). See *Figure 4—figure supplement 2* for an annotated version of this phylogeny that also includes the *acrs* recently discovered by Uribe et al (*acrIIA7 – acrIIA10*). (B) An unrooted phylogenetic tree of predicted AcrIIA11 homologs clusters into three major groups that correspond to the bacterial families depicted in (A) and *Figure 4—figure supplement 2*. Circles at nodes indicate bootstrap support >0.75. The star indicates the original AcrIIA11 sequence isolated via functional selection. Open and filled circles indicate whether a given homolog inhibits SpyCas9 in a plasmid protection assay. Untested homologs are indicated with a double-dagger (‡); WP_ 103240931.1 is shown to give

*Figure 4 continued on next page*

*Figure 4 continued*

context to *Figure 4—figure supplement 3A*. (C) Examples of MGEs that encode AcrIIA11 homologs from each group in (B); AcrIIA11 homologs from each group are found in a variety of MGEs (*Figure 4—figure supplement 3A*). AcrAIIA11 homologs are shown in black, other genes are colored by functional category. Accession numbers denote NCBI Genbank IDs and the genome coordinates bounding each locus are adjacent to each sequence. (D) AcrIIA11 homologs antagonize SpyCas9. Open circles depict statistically significant differences in plasmid retention between SpyCas9-inducing and non-inducing conditions (Student's t-test, p<0.01, n = 3). Closed circles depict samples for which SpyCas9 induction caused no significant difference in plasmid retention (Student's t-test p>0.05, n = 3). These symbols correspond to the node labels in (B). All p-values were corrected for multiple hypotheses using Bonferroni's method. Toxicity upon AcrIIA11c.4 expression prevented it from yielding meaningful data. Error bars depict standard error of the mean.

DOI: https://doi.org/10.7554/eLife.46540.012

The following figure supplements are available for figure 4:

**Figure supplement 1.** Summary statistics for AcrIIA11 homologs.

DOI: https://doi.org/10.7554/eLife.46540.013

**Figure supplement 2.** *AcrIIA11* is widely distributed across Firmicutes and related phyla.

DOI: https://doi.org/10.7554/eLife.46540.014

**Figure supplement 3.** *AcrIIA11* moves by horizontal gene transfer and is found nearby putative anti-CRISPR associated genes.

DOI: https://doi.org/10.7554/eLife.46540.015

**Figure supplement 4.** Diverse AcrIIA11 homologs inhibit TdeCas9.

DOI: https://doi.org/10.7554/eLife.46540.016

amino acid identity with SpyCas9. Intrigued by this observation, we tested divergent AcrIIA11 homologs from all three phylogenetic groups against SpyCas9 and found that homologs from each group could inhibit its activity (*Figure 4D*).

Because diverse AcrIIA11 homologs inhibit SpyCas9, we hypothesized that AcrIIA11 may intrinsically possess the capacity to antagonize a broad set of Cas9 orthologs. To test this hypothesis, we asked whether a panel of AcrIIA11 homologs could also inhibit the type II-A Cas9 protein from *Trepenoma denticola* (TdeCas9, 30/42% amino acid identity to SpyCas9/CqCas9) in a plasmid protection assay. Consistent with our predictions, we found that many homologs of AcrIIA11 can inhibit TdeCas9 (*Figure 4—figure supplement 4*). This indicates that AcrIIA11 can inhibit divergent Cas9 proteins, as has been observed for other Acrs (*Harrington et al., 2017*; *Lee et al., 2018*; *Marshall et al., 2018*). Because homologs of AcrIIA11 are found in many genera prevalent within human gut microbiomes (*Figure 4—figure supplement 2*), its broad inhibitory range suggests that AcrIIA11 homologs will pose a meaningful barrier to Cas9 activity in this habitat – both in the context of natural phage infections as well as Cas9-based interventions to manipulate microbiome composition (*Sheth et al., 2016*; *Pursey et al., 2018*).

Type II-A CRISPR-Cas systems are enriched in the *Lachnospiraceae* relative to other bacterial families (*Figure 4—figure supplement 2*, p=5×10$^{-27}$, $\chi^2$ test), which suggests that MGEs in these bacteria regularly encounter type II-A systems. This could explain why all tested AcrIIA11 homologs in the 'a' group (*Figure 4B*) inhibited SpyCas9 (*Figure 4D*). Consistent with their anti-CRISPR activity, two genes from this group, *acrIIA11a.1* and *acrIIA11a.2*, are found within 2 Kb of a putative *aca4* homolog (*Figure 4—figure supplement 3A*), which has been previously associated with multiple *acr* loci (*Marino et al., 2018*). Because *aca*-linked loci often encode hot spots of multiple *acr*s (*Pawluk et al., 2016b*; *Borges et al., 2017*; *Pawluk et al., 2017a*), we also tested 17 genes from these loci for their capacity to inhibit SpyCas9 in a plasmid protection assay. We found that none of these genes encoded strong inhibitors of SpyCas9 (*Figure 4—figure supplement 3B and C*), although it remains possible that they exhibit anti-CRISPR activity – perhaps against type I-C or III-A systems, as many of the tested genes are found in prophages which encounter these CRISPR-Cas subtypes (*e.g.* in *Ruminococcus_B lactaris*, NCBI accession QSQN01).

## A novel mode of SpyCas9 antagonism by AcrIIA11

Considering their relatively recent discovery, it is not surprising that very few Acrs have been mechanistically characterized. Nonetheless, a common theme has emerged from biochemical studies of AcrIIA2 and AcrIIA4, the only type II-A Acrs for which a mechanism of Cas9 inhibition has been elucidated. These studies have shown that both AcrIIA2 and AcrIIA4 are dsDNA mimics; they inhibit SpyCas9 by binding to its gRNA-loaded form and preventing association with a dsDNA target

(*Dong et al., 2017*; *Shin et al., 2017*; *Yang and Patel, 2017*; *Jiang et al., 2019*; *Liu et al., 2019*). Given that AcrIIA11 is an unrelated inhibitor, we sought to address whether it has the same mode of SpyCas9 antagonism as these previously-studied type II-A Acr proteins.

We therefore purified recombinant AcrIIA11 from *E. coli* to analyze its biochemistry, interactions with SpyCas9, and possible modes of inhibition. We first observed that untagged AcrIIA11 migrates at approximately twice its predicted molecular weight by size exclusion chromatography (*Figure 5A*), suggesting that it may function naturally as a dimer. Consistent with its ability to protect plasmids and phage from SpyCas9 in vivo, purified AcrIIA11 inhibited the dsDNA cleavage activity of SpyCas9 in-vitro, in a concentration-dependent manner (*Figure 5B and C*). These data demonstrate that AcrIIA11 is sufficient for SpyCas9 inhibition and does not require additional host or phage factors, at least in vitro. This autonomous activity is consistent with *acrIIA11*'s phylogenomic signature, as it shows no linkage to any other gene across phage genomes (*Figure 2—figure supplement 2*, *Figure 4—figure supplement 3A*).

We considered several possibilities for AcrIIA11's mode of antagonism. AcrIIA11 did not prevent SpyCas9 from binding a single-guide RNA (sgRNA) and we did not observe a strong AcrIIA11/sgRNA interaction (*Figure 5—figure supplement 1*). Notably, AcrIIA11 did limit migration of the SpyCas9/sgRNA ribonucleoprotein through a native gel, suggesting that AcrIIA11 may bind this complex (*Figure 5—figure supplement 1*). So, we next tested whether AcrIIA11 directly binds SpyCas9 (*Figure 5D*). We found that AcrIIA11 was unable to bind the I-SmaMI meganuclease (used as a negative control) but bound both the apo (without sgRNA) and sgRNA-loaded forms of SpyCas9. The addition of sgRNA enhanced AcrIA11 binding to SpyCas9 (*Figure 5D*, *Figure 5—figure supplement 2*) but to a lesser extent than previously documented for both AcrIIA2 and AcrIIA4 (*Dong et al., 2017*; *Shin et al., 2017*; *Yang and Patel, 2017*; *Jiang et al., 2019*; *Liu et al., 2019*).

Given the precedents set by AcrIIA2 and AcrIIA4, we next tested whether AcrIIA11 acts as a DNA mimic to antagonize SpyCas9. Both AcrIIA2 and AcrIIA4 have the key property of preventing sgRNA-complexed SpyCas9 from binding target dsDNA (*Dong et al., 2017*; *Shin et al., 2017*; *Yang and Patel, 2017*; *Jiang et al., 2019*; *Liu et al., 2019*). To test whether AcrIIA11 functions similarly, we performed an electrophoretic mobility shift assay (EMSA) in which we tracked the migration of 6-FAM labeled dsDNA upon SpyCas9 binding. Consistent with previous work (*Dong et al., 2017*; *Shin et al., 2017*; *Yang and Patel, 2017*), we found that AcrIIA4 prevents the gel-shift caused by SpyCas9/sgRNA binding to its target dsDNA (*Figure 6A*, lane 3). In contrast, we observe that AcrIIA11 does not prevent this gel-shift, even at molar ratios that abolish SpyCas9 cleavage activity (compare lanes 2 through 5; *Figure 6A*).

Instead of preventing a SpyCas9-induced gel-shift, we see that AcrIIA11 creates the opposite effect, giving rise to a super-shifted SpyCas9/sgRNA/dsDNA ternary complex (lane 4, *Figure 6A*). A similar super-shift occurs using both short and long dsDNA substrates, which differ in the amount of dsDNA unprotected by SpyCas9's footprint and thus exposed to AcrIIA11 (*Figure 6A*, *Figure 6—figure supplement 1A*). Because the shorter substrate has minimal dsDNA overhangs (five bp), AcrIIA11 binding to adjacent dsDNA is not likely to result in the super-shift observed; instead, we conclude that AcrIIA11 binds SpyCas9 to trigger this super-shift (*Figure 6B*). Consistent with this model, we observe that AcrIIA11 retards the migration of apo-, sgRNA-loaded, and dsDNA-bound SpyCas9 through a native gel in a concentration-dependent manner (*Figure 6C*, *Figure 6—figure supplement 1B and C*). These data suggest that AcrIIA11 binds a motif on SpyCas9 preserved across all three conformations of the enzyme. This also distinguishes AcrIIA11 from AcrIIA2 and AcrIIA4, which bind to the dsDNA binding site of SpyCas9 only available in its sgRNA-loaded form (*Dong et al., 2017*; *Shin et al., 2017*; *Yang and Patel, 2017*; *Jiang et al., 2019*; *Liu et al., 2019*).

In addition to binding SpyCas9, AcrIIA11 binds dsDNA weakly in the absence of SpyCas9 (lanes 8 and 9, *Figure 6A*), but strongly in the presence of either apo- or sgRNA-loaded SpyCas9 (compare 'band #2' in lanes 4 and 12 with 8 and 11, *Figure 6A*). We verified this behavior by Western blot, demonstrating that SpyCas9 does not co-migrate with 'band #2', thereby confirming that AcrIIA11 binds dsDNA to generate this gel-shift (*Figure 6C*, *Figure 6—figure supplement 1A and B*). Based on these findings, we hypothesize that AcrIIA11 undergoes a conformational change that enhances its dsDNA-binding upon interaction with SpyCas9. Consistent with this hypothesis, a Coomassie-stained native gel shows that AcrIIA11 forms a discrete, bright band in the presence of SpyCas9 but is more diffuse in its absence (*Figure 6—figure supplement 1D*), indicating that SpyCas9 may

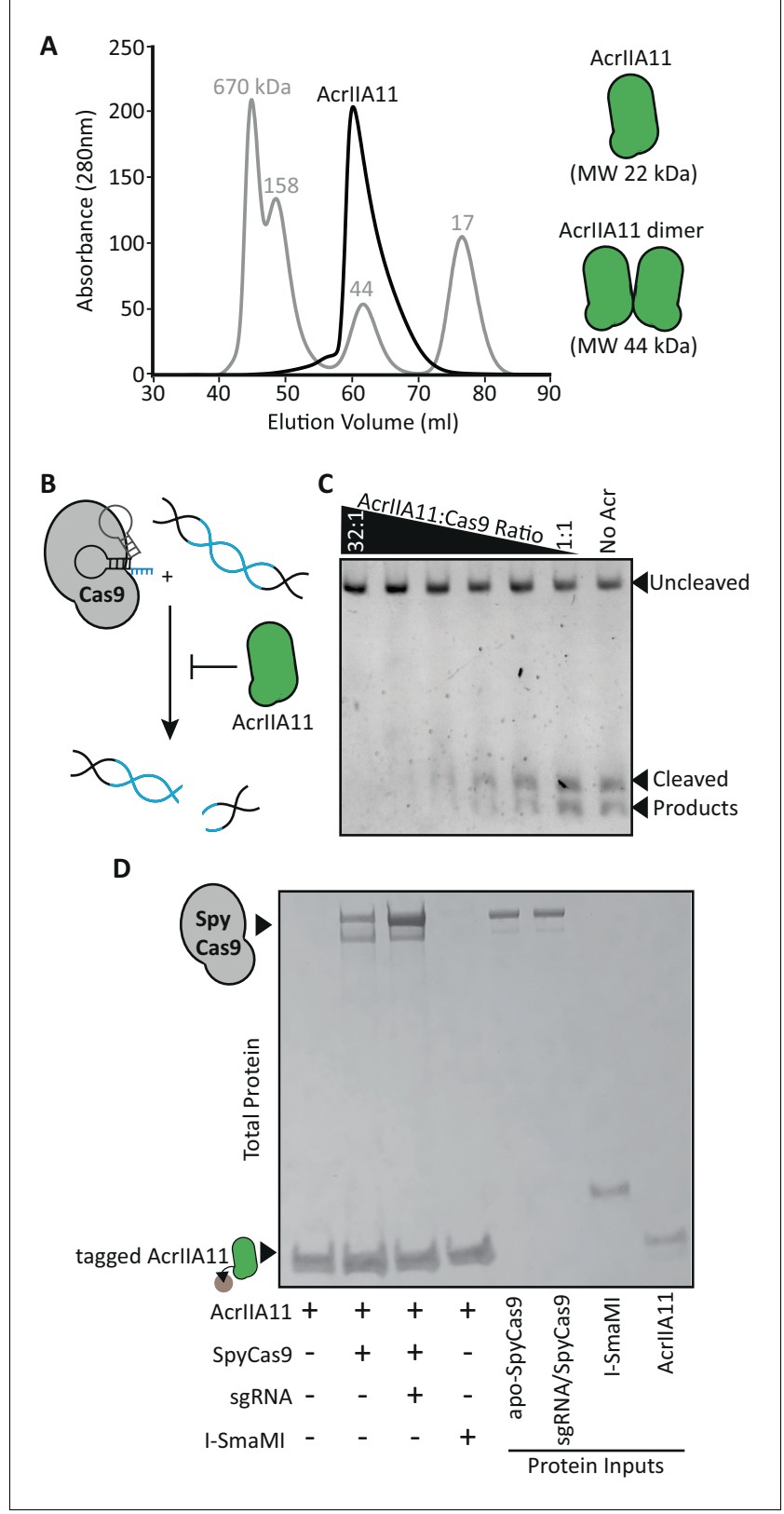

**Figure 5.** AcrIIA11 binds SpyCas9 to inhibit dsDNA cleavage. (**A**) AcrIIA11 elutes as a dimer on size exclusion chromatography. The gray trace depicts protein standards of the indicated molecular weight. The black trace shows AcrIIA11 elution. The predicted molecular weights of its monomeric and dimeric forms are depicted to the

*Figure 5 continued on next page*

*Figure 5 continued*

right. (**B**) Schematic of dsDNA cleavage assay in (**C**). (**C**) AcrIIA11 inhibits the ability of SpyCas9 (0.4 µM) to cleave a linear 2.6 Kb dsDNA substrate in a concentration-dependent manner. (**D**) AcrIIA11 binds SpyCas9, with a moderate preference for the sgRNA-loaded form. A Coomassie stain of total protein following pulldown of a 2x-strep-tagged AcrIIA11 incubated with either purified SpyCas9 (+ /- sgRNA) or the meganuclease I-SmaMI (as a negative control). SpyCas9 often runs as a doublet, likely due to partial protein degradation, and sgRNA does not influence SpyCas9 stability in these conditions (see protein inputs).
DOI: https://doi.org/10.7554/eLife.46540.017
The following figure supplements are available for figure 5:

**Figure supplement 1.** AcrIIA11 does not prevent a sgRNA gel-shift, indicating that the RNA is protein-bound.
DOI: https://doi.org/10.7554/eLife.46540.018
**Figure supplement 2.** AcrIIA11 binds SpyCas9 with a moderate preference for the sgRNA-loaded form.
DOI: https://doi.org/10.7554/eLife.46540.019

---

stabilize a conformation of AcrIIA11. In contrast to our result with dsDNA, we do not detect binding of AcrIIA11 to single-stranded DNA (*Figure 6—figure supplement 1A*).

In summary, we demonstrate that AcrIIA11 can directly bind multiple conformations of SpyCas9 and that SpyCas9 induces AcrIIA11 to bind dsDNA. Though our results leave uncertain which of these behaviors, in isolation or combination, contribute to SpyCas9 antagonism, they make clear that AcrIIA11 uses a different mechanism of inhibiting SpyCas9 than the dsDNA mimicry employed by AcrIIA2 or AcrIIA4.

## AcrIIA11 exhibits locus-specific inhibition of SpyCas9 in human cells

Because AcrIIA11 functioned in-vitro and in bacteria, we next sought to determine if it inhibited Spy-Cas9 activity in mammalian cells, which has been shown for a short list of other Acrs (*Rauch et al., 2017*; *Hynes et al., 2018*). We therefore transfected HEK293T cells with plasmids expressing Spy-Cas9 and either a genome-targeting or non-targeting sgRNA. In addition, we co-transfected a second plasmid expressing AcrIIA4 (as a positive control) or either of two AcrIIA11 homologs (AcrIIA11a.1 or AcrIIA11b.1). We first tested if these Acrs inhibited SpyCas9 cleavage at the *CAC-NA1D* target locus. As expected, we found that SpyCas9 was able to robustly cleave the *CACNA1D* target locus in the presence of the cognate sgRNA. Additionally, AcrIIA4 was able to potently block this SpyCas9 activity, consistent with previous studies (*Rauch et al., 2017*; *Shin et al., 2017*). We also found that AcrIIA11a.1, but not AcrIIA11b.1, inhibited SpyCas9's ability to generate genomic lesions as potently as AcrIIA4, (*Figure 7A and B*, *Figure 7—figure supplement 1*), consistent with the results from our plasmid protection assays (*Figure 4D*). These findings demonstrate that AcrIIA11 homologs can inhibit SpyCas9 in human cells.

Next, we asked whether AcrIIA11's inhibition of SpyCas9 was universal or locus-specific. We examined SpyCas9 cleavage at a second target locus: the *EMX1* gene (*Figure 7C and D*). As expected, we found that SpyCas9 robustly cleaves the *EMX1* target in the presence of the cognate sgRNA. Like with the *CACNA1D* locus, *EMX1* cleavage was also strongly inhibited by AcrIIA4. In contrast, we found that neither AcrIIA11 homolog could inhibit SpyCas9 activity at the *EMX1* target locus. These results further support our in vitro findings that the mechanism of AcrIIA4 and AcrIIA11 are substantially different. For example, AcrIIA4 binds SpyCas9 to prevent DNA target recognition, and can therefore inhibit SpyCas9 at all loci tested (*Figure 7*). In contrast, AcrIIA11 homologs can robustly inhibit SpyCas9 only at some loci. This indicates that some feature of the target site (for instance, its chromatin state) might impact AcrIIA11's activity.

## Discussion

The identification and characterization of CRISPR-Cas systems has outpaced the discovery of anti-CRISPRs, even though Acr diversity almost certainly exceeds CRISPR-Cas diversity in nature (*Pawluk et al., 2017a*; *Watters et al., 2018*). These undiscovered Acrs are likely to be important influences on both phage infection outcomes and horizontal gene transfer in natural microbial communities (*Borges et al., 2017*). Determining the impact of Acrs on these processes is a challenge, however, because *acrs* lack shared sequence features and so are difficult to detect. To overcome

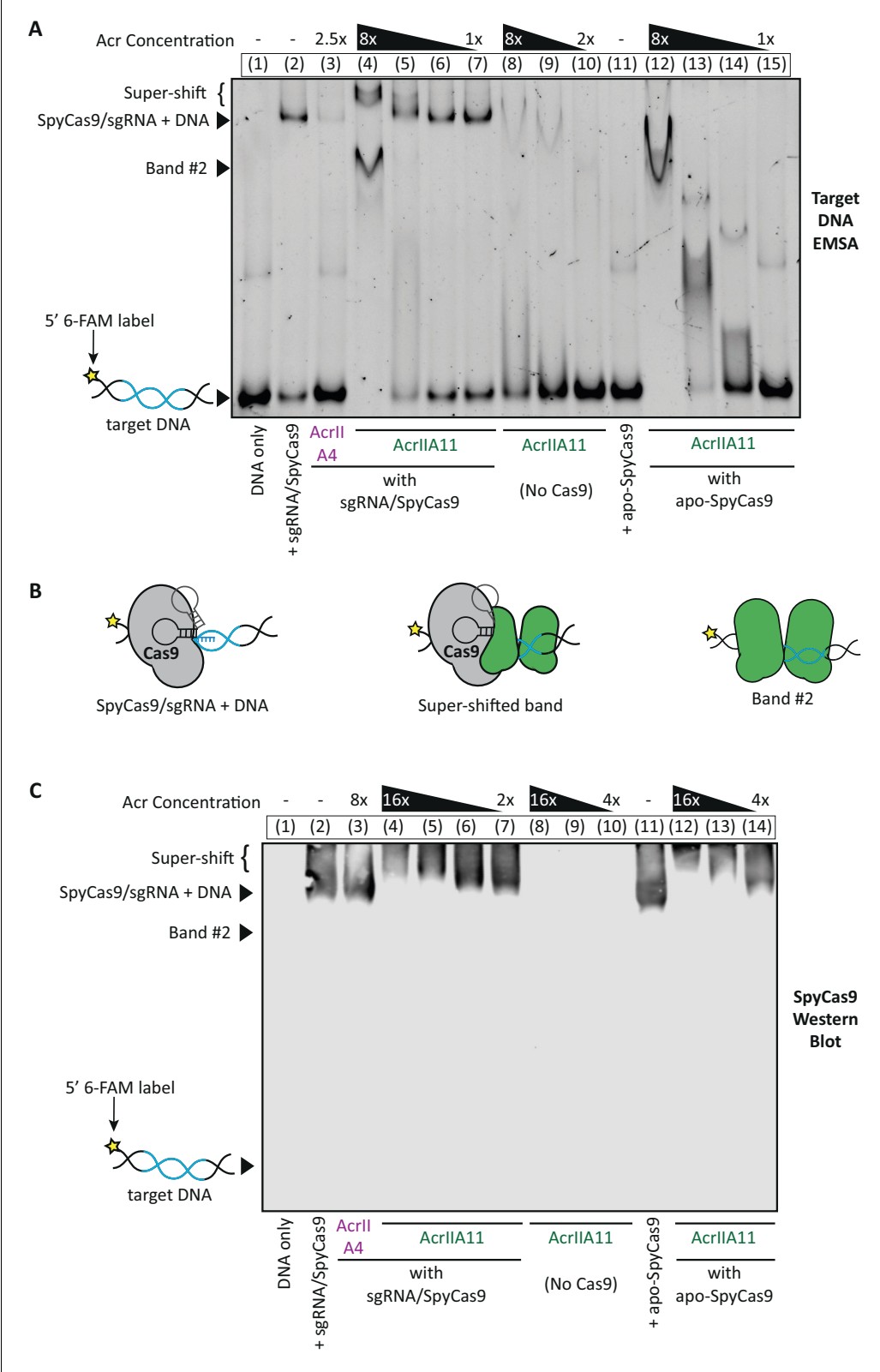

**Figure 6.** AcrIIA11 inhibits SpyCas9 via a novel mechanism. (**A**) Using an EMSA, we examined the gel shift experienced by a 60 bp target DNA, fluorescently labeled with a 6-FAM marker and incubated with purified SpyCas9, sgRNA, and/or purified Acrs. Divalent cations were omitted from reactions such that SpyCas9 could bind, but not cleave, target DNA, as in *Lee et al. (2018)*. Lane numbers and Acr concentrations (relative to SpyCas9) are depicted above the native gel. Lane contents are depicted below the gel; 2 μM SpyCas9 was used. Key bands are annotated to the left of the gel. *Figure 6 continued on next page*

Figure 6 continued

(B) Proposed explanations for the key bands in (A) are schematized. (C) A Western blot for SpyCas9 was performed on an EMSA like the one shown in (A). SpyCas9 does not co-migrate with band#2, indicating that AcrIIA11 binds dsDNA to form this band. The positions of unbound target DNA, band #2, SpyCas9-bound DNA, and the super-shift are indicated to the left of the blot. The positions of these key bands were determined using a matched DNA EMSA on a 36 bp target sequence, which is depicted alongside this blot in *Figure 6—figure supplement 1*. Taken together, (A) and (C) indicate that AcrIIA11 binds the SpyCa9/sgRNA/dsDNA ternary complex and is stimulated to bind dsDNA in the presence of SpyCas9.

DOI: https://doi.org/10.7554/eLife.46540.020

The following figure supplement is available for figure 6:

**Figure supplement 1.** AcrIIA11 inhibits SpyCas9 via a novel mechanism.
DOI: https://doi.org/10.7554/eLife.46540.021

this obstacle and enable function-based *acr* discovery, we developed a high-throughput selection that identifies *acr*s based solely on their activity. Applying our functional selection to human fecal and oral metagenomes, we recovered 51 DNA fragments that putatively antagonize SpyCas9, with 10 confirmed for activity; many additional host- or phage-encoded inhibitors are expected to be among the remaining 41 sequences.

In parallel work to ours, Uribe and colleagues recently described a functional metagenomic scheme to identify *acr*s (*Uribe et al., 2019*). They used a single round of Cas9 selection and one targeting crRNA to identify putative SpyCas9-antagonizing clones, though the authors suspected many of these to be false positives. To find *acr*s, they selected 39 ideal candidates from their initial dataset for re-testing, confirming 11 DNA fragments for anti-Cas9 activity, from which they identified four new *acr*s (*acrIIA7-10*). Although the premise of both of our approaches is similar, our strategy uses two iterations of SpyCas9 selection and two targeting crRNAs to identify *acr*s, which suppresses both Cas9 loss-of-function mutations and escape mutations in the target plasmid (*Figure 1D*, *Figure 1—figure supplement 1*, *Figure 1—figure supplement 2*). As a result, we confirmed that 10 of the 14 most abundant clones following selection could antagonize SpyCas9 (*Figure 2B*). More generally, we show that iterative selection can enable high-confidence discoveries from enormously diverse input libraries, which will be critical in the search for even rarer classes of *acr*s and other phage counter-defense strategies.

One of the major lessons from our work and that of Uribe et al is that the vast majority of *acr*s in nature remain undiscovered. Neither study encountered any previously described *acr*s and no *acr*s were shared across datasets, though more than $10^7$ unique clones were screened for antagonists of the same Cas9 allele (SpyCas9) in the same selection host (*E. coli*). This lack of overlap emphasizes that the few *acr*s which have been catalogued are a very minor subset of those that are found in nature. This tremendous diversity exists because selection favors a highly-varied repertoire of *acr*s among even related MGEs (*Bondy-Denomy et al., 2013*; *Borges et al., 2017*) and because Acrs have evolved independently many times from a variety of progenitor proteins (*Rollins et al., 2018*; *Stone et al., 2018*; *Uribe et al., 2019*). With extreme diversity favored, and the relative ease by which new Acr function is born, it is almost certain that much more mechanistic variety among Acrs remains to be discovered.

In this study, we have focused on the most potent SpyCas9 antagonist from our set — AcrIIA11 — extensively characterizing its phylogenetic distribution, functional range, and mechanism. SpyCas9 is unlikely to be the natural target of AcrIIA11, as it is quite divergent from the Cas9 homologs found in AcrIIA11-encoding bacteria (*Chylinski et al., 2014*). Yet, diverse AcrIIA11 homologs, which are distributed across multiple phyla, retain inhibitory activity against SpyCas9. This suggests that AcrIIA11 is intrinsically predisposed to act against a wide variety of Cas9 sequences, a prediction we verified by confirming that many AcrIIA11 homologs can inhibit a highly-diverged Cas9 ortholog from *T. denticola* (*Figure 4—figure supplement 4*). Such inhibitory breadth may be particularly useful to MGEs that infect bacteria with diverse CRISPR-Cas systems (*Makarova et al., 2015*; *Crawley et al., 2018*) and could explain AcrIIA11's broad phylogenetic distribution. This combination of prevalence and inhibitory breadth may also impact many Cas9-based interventions (*Sheth et al., 2016*; *Pursey et al., 2018*), especially in the gut microbiome, where AcrIIA11 homologs are particularly common.

Broad inhibition implies that AcrIIA11 might interact with conserved residues on Cas9. There is precedent for such biochemical activity in an Acr. For example, AcrIIC1, a broad-spectrum inhibitor

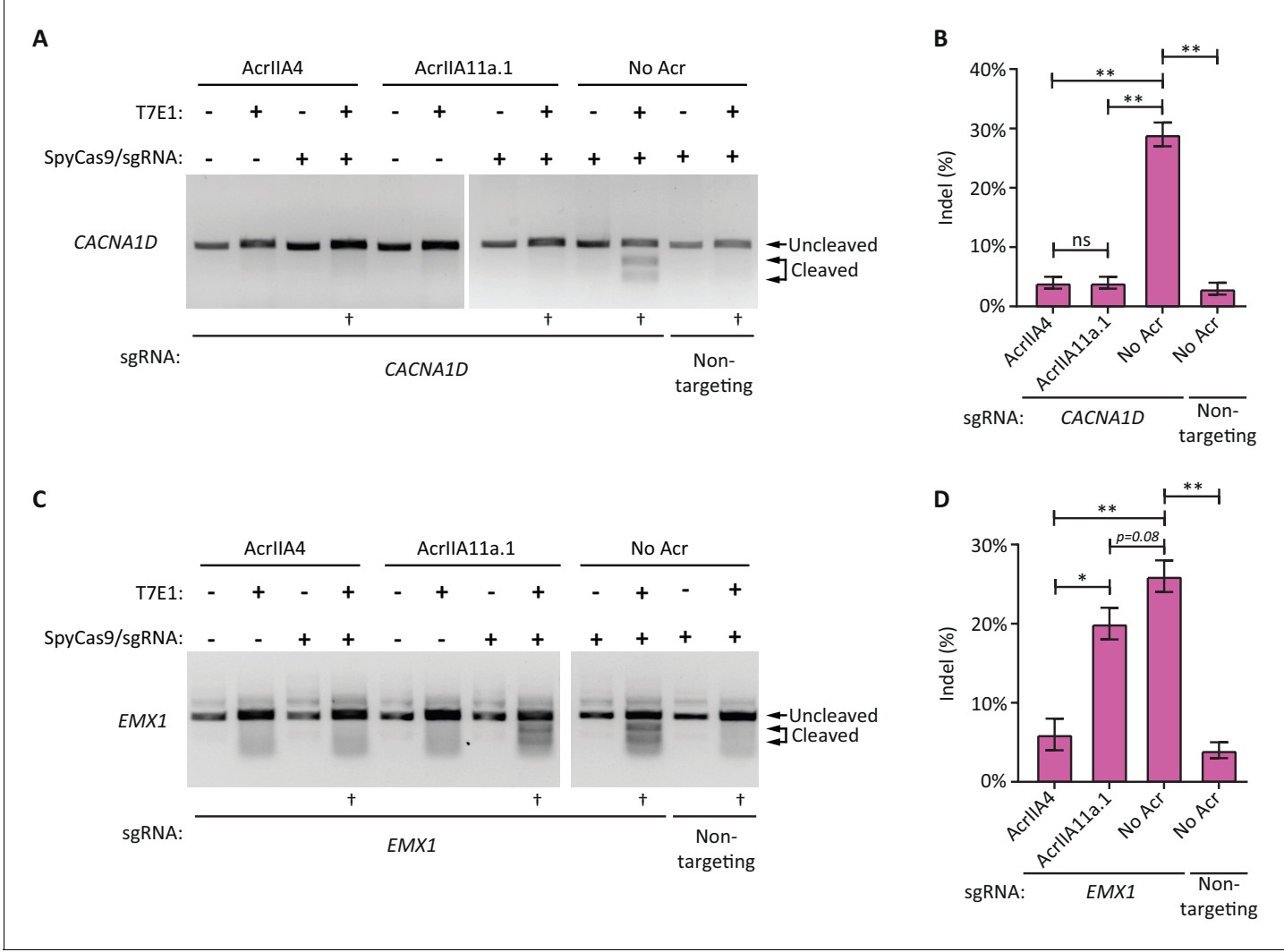

**Figure 7.** AcrIIA11 inhibits SpyCas9 activity in human cells. (**A**) AcrIIA4 and AcrIIA11a.1 inhibit SpyCas9 cleavage at the *CACNA1D* locus, as determined via a surveyor nuclease assay with T7 endonuclease I (T7E1). T7E1 cleaves dsDNA that has small insertions and deletions (indels) which result from SpyCas9-induced dsDNA breaks repaired via non-homologous end joining. This allowed for the quantification of SpyCas9 editing efficiency following transient transfection of HEK293T cells. For each experiment, the dagger (†) indicates one of three biological replicates transfected and treated with T7E1 to generate the data in (**B**). (**B**) Quantification of indel frequencies at the *CACNA1D* locus. Asterisks depict statistically significant differences in indel frequency (Student's t-Test, n = 3 biological replicates). (**C**) A representative gel image from a single T7E1 assay depicting SpyCas9 cleavage at the *EMX1* locus; the dagger (†) indicates samples used to generate the data depicted in (**D**). (**D**) Indel frequencies at the *EMX1* locus, as in (**B**). Double asterisks (**), p<0.001; Single asterisk (*), p<0.01; ns, not significant. All p-values were corrected for multiple hypotheses using Bonferroni's method.
DOI: https://doi.org/10.7554/eLife.46540.022

The following figure supplement is available for figure 7:

**Figure supplement 1.** AcrIIA11b.1 does not inhibit SpyCas9 in human cells.
DOI: https://doi.org/10.7554/eLife.46540.023

of type II-C Cas9 enzymes, binds to conserved Cas9 residues to prevent nuclease activity but not target DNA binding (*Harrington et al., 2017*). The same outcome is achieved by AcrIE1 and AcrIF3, two inhibitors of type I CRISPR-Cas systems, as each Acr impairs Cas3 nuclease activity but does not affect target DNA recognition (*Bondy-Denomy et al., 2015*; *Pawluk et al., 2017b*). No type II-A Cas9 inhibitor has been shown to act similarly, though AcrIIA11's behavior in-vitro is consistent with such a mechanism; it binds SpyCas9, inhibits DNA cleavage, but does not prevent target recognition.

Alternatively, AcrIIA11's inhibition of SpyCas9 could be related to dsDNA-binding ability. Though other Acrs can bind nucleic acids, such as AcrIIA1 and AcrIIA6 (*Hynes et al., 2018*; *Ka et al., 2018*), AcrIIA11 is the first Acr reported to bind both dsDNA and its target Cas protein (*Stanley and Maxwell, 2018*; *Trasanidou et al., 2019*). Furthermore, interaction with SpyCas9 dramatically improves AcrIIA11's dsDNA binding, linking these key behaviors. A role for dsDNA-binding in AcrIIA11's mode of inhibition could explain why it inhibited SpyCas9 at only some genomic loci (*Figure 7*). Chromatin state (or other locus-specific factors) would alter AcrIIA11's access different SpyCas9 target sites which, in turn, would influence its ability to inhibit SpyCas9 at these loci. In comparison, AcrIIA4 interacts with SpyCas9 but not target DNA and, consistent with this model, inhibited SpyCas9 at all tested target sites.

While AcrIIA11's exact mechanism still awaits elucidation, its mode of antagonism is clearly distinct from the other two mechanistically-characterized type II-A Acrs, AcrIIA2 and AcrIIA4, which act as dsDNA mimics to prevent target recognition (*Dong et al., 2017*; *Shin et al., 2017*; *Yang and Patel, 2017*; *Jiang et al., 2019*; *Liu et al., 2019*). A combination of Acrs that act via different mechanisms may be more effective at inhibiting CRISPR-Cas activity than a combination of Acrs that act redundantly (*Borges et al., 2017*). Thus, AcrIIA11's novel mode of antagonism could enable more potent SpyCas9 inhibition than can be achieved using only Acrs that act via dsDNA mimicry. AcrIIA11's effectiveness against SpyCas9 at some loci in human cells reinforces this possibility and highlights its potential use for modifying new Cas9-based tools in medicine and research.

Excitingly, AcrIIA11 represents just the tip of the iceberg. We have identified 51 metagenomic clones that putatively antagonize SpyCas9 (with 10 confirmed for activity); many of these sequences are likely to contain new SpyCas9 inhibitors, that is, ones not homologous to any known Acr. Functional metagenomics not only offers a powerful means to identify new *acr* genes, but also enables discovery of fundamentally new ways to inhibit Cas9. More broadly, our functional metagenomic approach allows us to detect *acrs* beyond curated sequence databases and can enable the discovery of Acrs against any CRISPR-Cas system of interest, in any microbial habitat, strain collection, or phage bank. This information will improve our understanding of phage and MGE dynamics in microbial ecosystems and holds significant promise for not only describing the longstanding evolutionary arms race between phages and bacteria, but also for improving phage- and CRISPR-Cas-based technologies used in therapeutic and engineering applications.

## Materials and methods

### Processing of metagenomic libraries

Metagenomic libraries were generously shared by Gautam Dantas (Washington Univ. in St. Louis) and have been described previously. Briefly, these libraries were constructed using oral swabs from Yanomami Amerindians (*Clemente et al., 2015*) or fecal samples from periurban residents of Lima, Peru (*Pehrsson et al., 2016*). These libraries contain 1.3–3.9 × $10^6$ unique clones with an average DNA insert size of 2 Kb (see *Supplementary file 1* table S2 for further details). To process and store each metagenomic library, the original freezer stock was inoculated into 52 ml of lysogeny broth (LB; 10 g/L casein peptone, 10 g/L NaCl, 5 g/L ultra-filtered yeast powder) and grown to an OD600 value of ~0.7 (300 µl/100 µl inoculum for oral/fecal libraries). All libraries were previously constructed in a pZE21 MCS1 plasmid backbone (henceforth, pZE21) marked by kanamycin resistance (Kan$^R$) (*Lutz and Bujard, 1997*; *Clemente et al., 2015*; *Pehrsson et al., 2016*). Each library was then titered on agar plates containing lysogeny broth with 50 ug/ml Kanamycin; 10–12 ml was used to create replicate freezer stocks. The remaining cells were pelleted by centrifugation at 4100 rcf for 6.5 min, purified using Qiagen miniprep kits (four columns per library), and quantified using the Qubit BR dsDNA quantification kit. The libraries Oral_3 and Oral_5 are combinations of two smaller libraries (*Clemente et al., 2015*); each smaller library was processed independently, the libraries combined in proportion to their number of unique clones, and then were transformed into *E. coli*.

### Cloning Cas9 expression constructs

Different plasmids (generically, pSpyCas9) were used to express *S. pyogenes* Cas9 during the development of the Acr selection scheme and follow-up work. Plasmid composition, applications, and resultant data are detailed in *Supplementary file 1* tables S1, S6, and *Figure 1—figure supplement*

*2*. Early versions of pSpyCas9 were built by modifying a previously described SpyCas9 expression vector (addgene #48645) (*Esvelt et al., 2013*) to target pZE21 with new crRNAs. This vector expresses SpyCas9, its trans-acting crRNA (tracrRNA), and all crRNAs from the same backbone. The targeting crRNAs were engineered into the original locus (*Esvelt et al., 2013*) by Gibson cloning with long-tailed primers. All Gibson assembly was performed using the NEBuilder HiFi DNA assembly mastermix [New England Biolabs (NEB) cat# E2621] per manufacturer's recommendations. The arabinose-inducible *araC* and pBad regulon was amplified from pU2 (*Lee et al., 2015*) and inserted into pSpyCas9 in place of the constitutive proC promoter using Gibson assembly to regulate SpyCas9 expression. For experiments using *T. denticola* Cas9 (TdeCas9), the *cas9* gene and tracrRNA locus from addgene plasmid #48648 were cloned into the same inducible vector. The crRNA locus was generated by two rounds of Gibson assembly using long-tailed primers: the first round generated the repeat sequence and the second round added the spacer indicated in *Supplementary file 1* table S6. In between these rounds, we noticed a frameshift in the *tdeCas9* gene, which we corrected using the Q5 site-directed mutagenesis kit (NEB cat #E0554). To create the first 2-crRNA pSpyCas9 construct, we ordered a gBlock from Integrated DNA Technologies (IDT) containing an I-SceI cut site and second crRNA (sequence crRNA 'Z' in *Supplementary file 1* table S6) under the control of pJ107106 and a T7TE-LuxIA terminator. This gBlock was cloned into pSpyCas9 by Gibson assembly. Swapping crRNA spacers at this locus was also performed by Gibson cloning with long-tailed primers. The pJ107106 promoter was converted to pJ107111 (*Zucca et al., 2015*) using the Q5 site-directed mutagenesis kit (NEB), with suggested protocols. All pSpyCas9 constructs were transformed into *Escherichia coli* (strain: NEB Turbo) by heat-shock and made electrocompetent as described (*New England Biolabs, 2015*).

## Metagenomic selection for SpyCas9 antagonists

In the first iteration of SpyCas9 selection, 360 ng of each metagenomic library was electroporated into a strain of ultra-competent *E. coli* (NEB Turbo) containing the spectinomycin-marked SpyCas9 expression plasmid (pSpyCas9). As a control, 360 ng of pZE21 expressing GFP was also transformed. We performed all electroporations in a 1 mM cuvette using a Bio-Rad Gene Pulser Xcell with the following settings: 2.1kV, 100 Ω, 25 μF. Following electroporation, cells recovered for three hours at 37°C in SOC media (NEB, cat. # B9020S). An aliquot was then taken to titer transformants and the remaining cells (860 μl) were inoculated into 25 ml of LB broth with 50 ug/ml spectinomycin (Spec) and 2 mg/ml arabinose. The GFP control was split across two 25 ml flasks, one with and one without arabinose (430 μl inoculum per flask). After 20 hr in selective conditions, all samples were titered on LB-Spec or LB-Kan/Spec plates. *Figure 1D* depicts the population proportion of Kan$^R$ colony forming units (cfu) at 20 hr after SpyCas9 induction relative to their proportion before SpyCas9 induction.

To isolate plasmids after one round of selection, colonies from LB-Kan/Spec titer plates (*Supplementary file 1* table S2) were collected in 3 ml of LB-Spec by scraping colonies with an L-shaped cell scraper (Fisher Scientific cat. # 03-392-151) to gently remove them from the agar. Colonies were collected from all metagenomic libraries and from the GFP control exposed to arabinose. One-third of the cells were used to make −80°C freezer stocks and plasmids from the remaining 2 ml were purified across two Qiagen miniprep columns into a combined 100 μl of nuclease-free H$_2$O. An I-SceI restriction site was engineered into pSpyCas9 to enable its removal via treatment with the homing endonuclease. For all samples, 51 μl of miniprep eluate was combined with 6 μl NEB cutsmart buffer and 3 μl (15 units) I-SceI, incubated for 20 hr at 37°C, and the reaction heat-killed at 65°C for 20 min. After withholding 5 μl of each sample for gel electrophoresis, 2.98 μl of the *E. coli* RecBCD enzyme (149 units, given generously by Andrew Taylor), 4.9 μl 10 mM ATP, and 1.62 μl NEB cutsmart buffer were added to the remaining 55 μl of each sample and incubated for one hour at 37°C. To stop the RecBCD reaction, EDTA was added to a final concentration of 20 mM and the sample incubated at 70°C for 30 min. Linearization of pSpyCas9 by I-SceI and the subsequent removal of linear DNA by RecBCD was confirmed by visualization with gel electrophoresis. Plasmid preparations were then purified through a Zymo Research DNA Clean and Concentrator column and the metagenomic DNA libraries electroporated into the original SpyCas9 selection strain a second time (*Supplementary file 1* table S2). For the second iteration of SpyCas9 selection, electroporation, recovery, induction, titering, and outgrowth was performed exactly as described for the first iteration.

## Purification and amplification of metagenomic DNA from SpyCas9 selections

From twice-selected metagenomic libraries, LB-Kan/Spec titer plates were used to collect Kan[R] clones from the population. Two titer plates were processed independently for each library and were chosen to have approximately 100 and 10,000 colonies. Only one titer plate was collected for the GFP control. For each plate, colonies were collected, stored, and plasmids then purified as described above. No enzymatic removal of pSpyCas9 was performed. Instead, purified plasmids were diluted ten-fold and metagenomic DNA fragments amplified via PCR using 45 μl Platinum HiFi polymerase mix (Thermo Fischer cat. # 12532016), 1.67 μl plasmid template, and 5 μl of a custom primer mix. The custom primer mix contained three forward and three reverse primers, each targeting the sequence immediately adjacent the metagenomic clone site in pZE21, staggered by one base pair. The stagger enabled diverse nucleotide composition during early Illumina sequencing cycles. A sample PCR contained the following primer volumes, each from a 10 μM stock: (primer F1, 5′-CCGAATTCATTAAAGAGGAGAAAG, 0.83 μl); (primer F2, 5′- CGAATTCATTAAAGAGGA-GAAAGG, 0.83 μl); (primer F3, 5′- GAATTCATTAAAGAGGAGAAAGGTAC, 0.83 μl); (primer R1, 5′-GATATCAAGCTTATCGATACCGTC, 0.36 μl); (primer R2, 5′- CGATATCAAGCTTATCGATACCG, 0.71 μl); (primer R3, 5′-TCGATATCAAGCTTATCGATACC, 1.43 μl). PCRs were then executed using the following conditions: 94°C for 2 min, 30 cycles of: 94°C for 10 s + 55°C for 30 s + 68°C for 5.5 min, and 68°C for 10 min. The amplified metagenomic fragments were cleaned using a Zymo Research DNA Clean and Concentrator column and eluted in 80 μl of Qiagen elution buffer.

## Illumina sample preparation and sequencing

For each sample, 50 μl of PCR amplicons were diluted to a final volume of 130 μl in Qiagen elution buffer and sheared to a fragment size of approximately 500 bp using a Covaris LE220 sonicator. Sheared DNA was purified and concentrated using a Zymo Research DNA Clean and Concentrator column and eluted in 25 μl nuclease-free $H_2O$. DNA was then end-repaired for 30 min at 25°C using 250 ng sample in a 25 μl reaction with 1.5 units T4 Polymerase (NEB cat. # M0203), five units T4 polynucleotide kinase (NEB cat. # M0201), 2.5 units Taq DNA polymerase (NEB cat. # M0267), 40 μM dNTPs, and 1x T4 DNA ligase buffer with 10 mM ATP (NEB cat. # B0202). Reactions were then heat-killed at 75°C for 20 min and 2.5 μl of 1 μM pre-annealed, barcoded sequencing adapters added with 0.8 μl of NEB T4 DNA ligase (cat. # M0202T). This reaction was incubated at 16°C for 40 min and heat-killed at 65°C for 10 min. The barcoded adapters contained a 7 bp sequence specific to each sample, which allowed mixed-sample sequencing runs to be de-multiplexed. Forward and reverse sequencing adapters were stored at 1 μM in TES buffer (10 mM Tris, 1 mM EDTA, 50 mM NaCl, pH 8.25) and annealed by heating to 95°C and cooling at a rate of 0.1 °C/sec to a final temperature of 4°C. Adapters were stored at −20°C and thawed slowly on ice before use. After adapter-ligation, samples were purified through a Zymo Research DNA Clean and Concentrator column and eluted in 12 μl of the supplied elution buffer.

Next, 6 μl from each sample was pooled together and the mixture size-selected to ~500–900 bp using a 1.9% agarose gel visualized with SYBR-Safe dye (Thermo Fisher cat. # S33102). DNA was purified from gel slices using a Zymo Research gel DNA recovery kit, eluting in 28 μl nuclease-free $H_2O$. Purified DNA was then enriched by PCR; a sample 25 μl reaction (in 1x Phusion HF buffer) contained 6 μl purified DNA template, 0.5 μl dNTPs (10 mM), 0.25 μl Phusion polymerase (two units/μl, Thermo Fisher cat. # F530), and 1.25 μl each of the following primers (10 μM stock): (primer F, 5′ – AATGATACGGCGACCACCGAGATCTACACTCTTTCCCTACACGACGCTCTTCCGATCT); (primer R, 5′ – CAAGCAGAAGACGGCATACGAGATCGGTCTCGGCATTCCTGCTGAACCGCTCTTCCGATCT). PCRs were amplified for 30 s at 98°C, subjected to 18 cycles of 98°C for 10 s, 65°C for 30 s, and 72°C for 30 s, and then incubated at 72°C for 5 min. Following enrichment PCR, samples were size-selected a second time to ~500–900 bp as previously described and purified libraries quantified using the Qubit fluorimeter (HS assay). Finally, 300 cycles of paired-end sequence data were generated at the Fred Hutchinson Cancer Research Center Genomics Core using a 17 pM loading concentration on an Illumina MiSeq machine with the v3 reagent kit.

## Assembly and annotation of functional metagenomic selections

Illumina paired-end reads were binned by barcode (perfect match required) such that data from each titer plate was assembled and annotated independently. Metagenomic DNA fragments from each sample were assembled using the first 93 bp of each read with PARFuMS, a tool developed specifically for assembling small-insert functional metagenomic selections and which was optimized for shorter read lengths (*Forsberg et al., 2012*). The PARFuMS assembler uses three iterations of Velvet (*Zerbino and Birney, 2008*) with variable job size, two iterations of PHRAP (*de la Bastide and McCombie, 2007*), and custom scripts to clean reads, remove assembly chimeras, and link contigs by coverage and shared annotation, as previously described (*Forsberg et al., 2012*). In sum, 206 contigs were assembled from all ten titer plates (encompassing five selections, *Figure 1—figure supplement 3*). We predicted open reading frames (ORFs) from each contig with MetaGeneMark (*Zhu et al., 2010*) using default parameters. Proteins were annotated by searching their amino acid sequences against the TIGRFAMs (*Haft et al., 2001*) and Pfam (*Bateman et al., 2000*) profile HMM databases using HMMER3 (*Finn et al., 2011*). The highest-scoring profile was used to annotate each ORF (minimum e-value $1e^{-5}$); these automated annotations were curated by hand into the functional groups shown in *Figure 2—figure supplement 1*.

The NCBI taxonomies of these contigs were determined using BLASTn against the 'nt' database; all contigs could confidently be assigned NCBI taxa at the resolution of order (*Figure 2—figure supplement 1*, *Supplementary file 1* table S4). Seven assembled contigs matched pSpyCas9 (>99% nucleotide identity) and were removed from the dataset. We also discarded contigs covered by less than two reads/base-pair/million reads (RPBM) to ensure contigs assembled from background DNA (*i.e.* plasmid backbone, host genome) were not considered (the coverage maximum among pSpyCas9-derived contigs was 1.67 RPBM, which informed our 2 RPBM cutoff). Then, contigs which could be linked to a mutation in either protospacer or protospacer adjacent motif (PAM) were eliminated (see Materials and methods section below). Note that we use the term 'target site' to refer to a particular protospacer/PAM sequence. Subsequently, redundant contigs assembled across both titer plates from a given library were removed with CD-HIT (*Li and Godzik, 2006*), using the following parameters: -c 0.95, -aS 0.95, -g 1 (the shorter contigs among clusters sharing 95% nucleotide identity over 95% the length of the shorter contig were removed). We checked for cross-contamination among the non-redundant contigs with a BLASTn search against the unfiltered contig set, detecting (and discarding) just one cross-contaminating contig (in library Fecal_01E). Manual curation of this near-final dataset prompted removal of one chimeric contig and the fusion of two incomplete assemblies. Note that the final dataset may exclude some metagenomic clones that inhibit SpyCas9, specifically those with mild antagonism or high fitness costs (rare among surviving colonies, with coverage <RPBM = 2) and those which have acquired target site mutations yet also encode SpyCas9 antagonists (partial antagonism may potentiate target site mutation). Indeed, one contig (O5_7) contains a mutation in the PAM of target site B yet still confers SpyCas9 protection when re-cloned into a wild-type vector backbone (*Figure 2B*). On the basis of empirical antagonism, we include this contig in our final dataset (to give 51 final contigs across five selections, *Supplementary file 1* tables S3, S4), but exclude all untested contigs linked to target site mutations. While these exclusions may omit *bona-fide* SpyCas9 antagonism, their exclusion yields a dataset with higher-confidence enrichment for SpyCas9 antagonists. All final assembled contigs have been submitted to GenBank under accession numbers MK637556 - MK637606 (*Supplementary file 1* table S4).

## Determining coverage of assembled contigs

Reads used as input for each assembly [i.e. those after adapter-trimming, vector-masking, and spike-in removal by PARFuMs (*Forsberg et al., 2012*) were also used to calculate coverage across each contig. For each sample, these reads were mapped to assembled contigs with bowtie2 (local alignment, unpaired reads, very-fast mode) and RPBM calculated for each contig.

## Determining the proportion of wild-type target sites

To estimate the proportion of wild-type target sites in each library after two rounds of SpyCas9 selection, sample-specific reads (before vector-cleaning) were concatenated by library and mapped to the pZE21 vector with bowtie2 (end-to-end alignment, unpaired reads, sensitive mode). Base frequencies at each protospacer and PAM position were calculated using bam-readcount (https://

github.com/genome/bam-readcount), requiring minimum mapping and base quality scores of 25 and 30, respectively. The proportion of wild-type target sites was conservatively calculated as the product of wild-type base calls at each protospacer/PAM position. In two libraries, (Oral_3 and Fecal_01E), sanger sequencing of individually picked colonies revealed a recombination event between a metagenomic DNA fragment and target site B, which would remove this SpyCas9 target from these clones (but avoid a frameshift within the kanamycin resistance gene). For these two libraries, reads were mapped to a variant of the pZE21 backbone corresponding to each recombination mutant. The proportion of reads mapping across the recombination site was compared to the proportion mapping to the wild-type vector backbone to estimate the frequency of recombination mutants in the population (0.4% and 24.2% for Oral_3 and Fecal_01E, respectively). The overall proportion of wild-type target sites in these libraries was then adjusted to reflect the proportion of these recombinants (*Supplementary file 1* table S3).

## Linking contigs to target site genotype

For some of the most abundant contigs in the final dataset, the genotype of the target site sequences in the corresponding plasmid backbone could be determined by Sanger sequencing individual Kan^R clones following two rounds of SpyCas9 selection. To link additional contigs with target site genotype in the clones not sampled by our Sanger data, we examined paired-end reads that mapped to target site sequences and an assembled contig. Because we used PCR to amplify metagenomic fragments before Illumina library creation, only a small minority of reads mapped to the vector backbone. Additionally, target sites A and B are 514 bp and 365 bp from the clone site of pZE21, respectively, so only read pairs from long sequencing inserts could link target site and contig genotype. Accordingly, many assembled contigs remain without links to target site genotype (*Figure 2—figure supplement 1*), though the overall proportion of wild-type target sites serves as a proxy for the number of SpyCas9 antagonists versus escape mutants in a given library (*Supplementary file 1* table S3).

For each titer plate, paired-end reads were mapped to a reference database containing all assembled contigs in both forward and reverse orientations within the pZE21 plasmid. We used bowtie2 in 'no-mixed' mode to consider only paired reads with an insert size ≥515 bp when mapping against this reference database (additional mapping options: end-to-end alignment, paired reads, sensitive mode). Only reads with a mapping quality score over ten were considered. We determined the true orientation of a metagenomic DNA fragment by counting whether more reads mapped across the clone-site junctions of the contig in the forward or reverse orientation. If more than one mapped read supported a particular protospacer/PAM variant, the contig linked to that target site was removed from the final dataset. Similarly, if fewer than ten reads mapped to a target site, a single read with any deviation from a wild-type protospacer/PAM was sufficient to eliminate the corresponding contig from further consideration.

## Predicting phage-associated contigs

The assembled metagenomic contigs (N50 length 2.2 Kb) were too short to provide sufficient information content for phage prediction based on their nucleotide sequence (*Roux et al., 2015*). Instead, protein sequences were used: those with close relatives in predicted phages were used to classify their parent contig as phage-associated. Predicted proteins from each contig were queried against NCBI's NR database using BLASTp (on September 12th, 2017) and the top five unique hits retained from each query. For each BLAST hit, we downloaded sequence 25 Kb upstream and downstream of the corresponding gene sequence and used these ~50 KB sequences as input for phage prediction with VirSorter (*Roux et al., 2015*). Sequences were compared against VirSorter's more encompassing 'virome' database (using default parameters) and hits to bacteriophage of any confidence category (from 'possible' to 'most confident') were considered phage-associated (*Roux et al., 2015*). The proteins that seeded these bacteriophage predictions were used then to classify metagenomic DNA contigs. If any predicted protein from our assembled contigs had a top-five BLAST hit found in a predicted bacteriophage, the whole metagenomic contig was classified as being phage-associated. These classifications are referenced in *Figure 2A* and *Supplementary file 1* table S4.

## Validating contigs with SpyCas9 protection

All contigs assembled from libraries Fecal_01A and Oral_5 with coverage above the sample median (RPBM >3.6, Fecal_01A and RPBM >8.2, Oral_5) were chosen for validation, totaling 15 contigs. Five contigs were represented in a sparsely-sampled set of individually-picked and Sanger-sequenced colonies. In these cases, the corresponding plasmids were re-transformed into NEB Turbo, target site sequences confirmed as wild-type by additional Sanger Sequencing, and the plasmids co-transformed with pSpyCas9 into NEB Turbo for functional testing. The remaining contigs were amplified by PCR from the plasmid preparations used for Illumina library construction. PCR primers (*Supplementary file 1* table S7) targeting the junction between pZE21 and the assembled contigs were used to facilitate re-cloning of the amplified fragments into pZE21 by Gibson assembly. PCRs used the high-fidelity Q5 polymerase (NEB) per suggested protocols. Cycling conditions are as follows: 98°C for 3 min, 35 cycles of 98°C for 15 s + 61°C or 65°C for 30 s + 72°C for 3 min, and 72°C for 10 min. Reaction-specific annealing temperatures are listed in *Supplementary file 1* table S7 and PCR products of the expected size were purified from gel slices using a Zymo Research gel DNA recovery kit. Nine of ten PCRs were successful [a contig from Fecal_01A (RPBM = 3.66) did not amplify well]. All Gibson assembly was performed using the NEBuilder HiFi DNA assembly master-mix per manufacturer's recommendations. Following Gibson-assembly, sequence-verified clones were co-transformed with pSpyCas9 into NEB Turbo, resulting in 14 total strains re-tested for Spy-Cas9 antagonism.

These 14 strains and a control carrying an empty pZE21 vector were grown overnight in LB with 50 µg/ml kanamycin and spectinomycin (LB-Kan/Spec) to maintain the pZE21 variants and pSpyCas9, respectively. The next morning, cultures were diluted 30-fold into LB-Kan/Spec and grown to log phase (absorbance readings at 600 nm ranged from 0.2 to 0.4). Mid-log cultures were then inoculated into LB with 50 ug/ml spectinomycin and either 0.2 mg/ml arabinose (to induce SpyCas9) or no arabinose. Kanamycin was omitted to allow for elimination of the pZE21 plasmid. Inocula were normalized by optical density (a 1:40 inoculum was used for an absorbance reading of 0.4). Cultures were then arrayed in triplicate across a 96-well plate (Greiner cat#655083, 200 µl per well) and grown in a BioTek Cytation three plate reader at 37°C with linear shaking at 1096 cycles per minute (cpm). After six hours of growth (at which time cultures had reached stationary phase), each well was serially-diluted ten-fold in peptone-NaCl (1 g/L each, pH 7.0). Spots (5 µl) of each dilution were plated on agar plates with LB-Kan/Spec (to count Kan$^R$ colonies) and LB-Spec (to count total colonies). *Figure 2B* depicts the log-transformed proportion of Kan$^R$/total cfu with and without SpyCas9 induction.

## Identification of causal Acr proteins

Null alleles of each predicted ORF in F01A_2 were created by inserting an early stop codon into each predicted protein. Stop codons were inserted by PCR using the high-fidelity Q5 polymerase (NEB) per suggested protocols with the following cycling conditions: 98°C for 3 min, 35 cycles of 98°C for 15 s + [annealing temperature] for 30 s + 72°C for [extension time], and 72°C for 10 min. Reaction-specific primers and conditions are listed in *Supplementary file 1* table S7. Constructs were then prepared using the Q5 site-directed mutagenesis kit (NEB) or by Gibson assembly (using the NEBuilder HiFi assembly mastermix, cat# E2621) per manufacturer's recommendations. Final constructs were sequence-verified, co-transformed with pSpyCas9 into NEB Turbo, and re-tested for SpyCas9 antagonism alongside an empty-vector control and the parent pZE21-F01A_2 construct. Plasmid protection assays were performed exactly as described in the section titled 'Validating contigs with SpyCas9 protection', except that SpyCas9 expression was induced using 2 mg/ml arabinose (rather than 0.2 mg/ml).

## Cloning candidate Acrs

The third ORF (*acrIIA11*) from F01A_2 and *acrIIA4* were codon-optimized for *E. coli* and sub-cloned into the KpnI (5') and HindIII (3') sites of pZE21 containing the *tetR* gene to allow for doxycycline-induced expression of the candidate Acrs from the pLtetO-1 promoter. To generate this plasmid, *tetR* (and its pLac promoter + rrnB terminator) was amplified from pCRT7 (addgene #52053) with NEB's Q5 polymerase per suggested protocols and using the conditions listed in *Supplementary file 1* table S7. This amplicon was then cloned by Gibson assembly into the pZE21

backbone. To clone *acrIIA4*, *acrIIA11*, and most of the candidate *acr*s near *acrIIA11a.1* (depicted in *Figure 4—figure supplement 3*), KpnI and HindIII sites were added to each gene by PCR with the Q5 polymerase (see *Supplementary file 1* table S7), vectors and genes digested using these enzymes, and sticky-end ligations performed using the Fast-Link ligation kit (epicentre cat# LK0750H) per suggested protocols. Genes flanking *acrIIA11a.1* were amplified from the F01A_2 contig, as this contig had proteins identical to those encoded by the genome depicted in *Figure 4—figure supplement 3* (one notable exception: the final 20 aa of *orf_d* in this figure were truncated in F01A_2 and a different 12 aa C-terminus was cloned, see *Supplementary file 1* table S8). This gene was cloned by Gibson assembly rather than KpnI/HindIII digests. The *acrIIA11 and acrIIA4* gene sequences were codon-optimized and synthesized as gBlocks from IDT (the AcrIIA4 amino acid sequence is identical to NCBI accession AEO04689.1). So were the candidate *acr* genes flanking *acrIIa11a.2* (depicted in *Figure 4—figure supplement 3*), which we cloned by Gibson assembly into the inducible pZE21_tetR backbone, using vector digested with KpnI and HindIII (via recommended protocols from NEB). AcrIIA11 homologs were codon-optimized for *E. coli* and synthesized by Gen-Script. They were also cloned into the pZE21_tetR vector. All constructs were co-transformed with Cas9-expressing plasmids (*Supplementary file 1* table S6) into NEB Turbo prior to performing plasmid protection assays. Because NEB Turbo contains the *laci^q* mutation, 0.5 mM IPTG was used throughout cloning and handling of the pZE21_tetR plasmid in this strain. IPTG was used because *TetR* is driven by the pLac promoter in this construct. Including IPTG ensured that TetR was at sufficient intracellular concentrations to maintain repressed transcription from the pLtetO-1 promoter prior to Acr induction (Acrs were expressed from the pLtetO-1 promoter).

## Testing candidate Acrs for plasmid protection

Overnight cultures of candidate Acr constructs were grown in LB-Kan/Spec + 0.5 mM IPTG (LB-Kan/Spec/IPTG) to maintain pCas9 and the Acr variants in an uninduced state. The next morning, cultures were diluted 1:50 into LB-Kan/Spec/IPTG and grown at 37°C for approximately two hours to mid-log with 100 ng/ml doxycycline to control Acr expression (final absorbance readings ranged from 0.2 to 0.6). In *Figure 3B*, doxycline was omitted from the indicated sample. Mid-log cultures were then inoculated into LB with 50 ug/ml spectinomycin, 0.5 mM IPTG, and either 2 mg/ml arabinose (to induce Cas9) or no arabinose. Doxycycline was included in this medium at 100 ng/ml to maintain Acr expression for previously induced cultures. Kanamycin was omitted to allow for elimination of the pZE21 target plasmid and inocula were normalized by optical density (a 1:40 inoculum was used for an absorbance reading of 0.4). Cultures were grown in a 96-well plate reader and the proportion of Kan^R cfu determined exactly as described in the section 'Validating contigs with SpyCas9 protection'. All figures depicting these data show the log-transformed proportion of Kan^R/total cfu, both with and without Cas9 induction, for each candidate Acr.

## Phage plaquing assay

Overnight cultures with pSpyCas9 and pZE21_tetR expressing either GFP or an Acr were diluted 1:50 in LB-Kan/Spec/IPTG supplemented with 5 mM MgSO$_4$ and grown shaking at 37C for three hours until late-log phase (OD600 range 0.5–0.8, see *Supplementary file 1* table S6 for exact crRNA sequences). GFP and the candidate Acrs were induced by adding doxycycline to a final concentration of 100 ng/ul and cultures grown for another two hours before SpyCas9 was induced by adding 0.2 mg/ml arabinose. After three additional hours of growth in the presence of both inducers, 200 µl of culture was used in a top-agar overlay, allowed to harden, and ten-fold serial dilutions of phage Mu spotted on top. The top and bottom agar media were made with LB-Kan/Spec/IPTG supplemented with 5 mM MgSO4, 0.02 mg/ml arabinose, and 100 ng/ul doxycycline and contained 0.5% and 1% Difco agar, respectively. Plates were incubated at 37°C overnight and plaques imaged the following day.

## AcrIIA11 homolog discovery and phylogenetic placement

To determine the distribution of AcrIIA1 - AcrIIA11 across bacteria (*Figure 4A* and *Figure 4—figure supplement 2*), homologs of each Acr were identified via a BLASTp search against NCBI nr; all sequences with ≥35% amino acid identity that cover ≥75% of the query length were retrieved. For AcrIIA11, these thresholds were roughly equivalent to an e-value threshold of $1e^{-50}$ following three

iterative psi-BLAST searches (*Figure 4—figure supplement 1*). We chose to use percent-identity and query-length thresholds to retrieve Acr homologs, rather than psi-BLAST e-values, because the position-specific scoring matrix used in psi-BLAST searches are not comparable across queries. We found that many AcrIIA11 homologs were derived from bacteria with phylogenetically-ambiguous taxonomic labels in NCBI. For instance, the NCBI genus 'Clostridium' is extremely polyphyletic, appearing in 121 genera and 29 families in a rigorous re-assessment of bacterial phylogeny performed by the genome taxonomy database, or GTDB (*Parks et al., 2018*). Importantly, the GTDB taxonomy ensures that taxonomic labels are linked to monophyletic groups, applies taxonomic ranks (phylum, class, order, etc.) at even phylogenetic depths, and substantially improves problematic taxonomies (*Parks et al., 2018*). To make phylogenetically meaningful inference with respect to AcrIIA11, we adopted the GTDB taxonomic scheme which, crucially, is largely congruent with NCBI taxonomy outside a few key problem areas (e.g. Clostridiales, see *Supplementary file 1* table S4). The taxonomic assignments for AcrIIA1-6 homologs are unchanged between NCBI and GTDB schema.

To map protein homologs onto a bacterial genus tree, the NCBI genome assemblies corresponding to each homolog were downloaded and classified according to the GTDB scheme using the 'classify_wf' workflow within GTDB toolkit (v0.1.3) (*Parks et al., 2018*); default parameters were used. To visualize these classifications on a tree of life, the minimal phylogenetic tree encompassing all lineages encoding AcrIIA1-6 or AcrIIA11 homologs was downloaded from AnnoTree (v1.0; node ID 31285). The AnnoTree phylogeny additionally includes KEGG and PFAM annotations for nearly 24,000 bacterial genome assemblies and is classified according to GTDB taxonomy (*Mendler et al., 2018*). Node 31285 includes four bacterial phyla which are identified with letters on *Figure 4—figure supplement 2* (a: Firmicutes, b: Firmicutes_D, c: Firmicutes_A, d: Firmicutes_F). The KEGG identifier K09952 was used to determine which GTDB genera encoded SpyCas9 while the PFAM IDs PF09711 and PF16813 were used to identify taxa which encode Csn2, the signature protein of type II-A CRISPR-Cas systems (*Makarova et al., 2015*). Enrichment for type II-A CRISPR-Cas systems was determined via a chi-squared test using the number of Csn2-encoding genomes within each GTDB family, with p-values corrected for multiple hypothesis testing using the Bonferroni method. Reciprocal best blast hits were used to assign homology to genes near AcrIIA11 loci (e.g. in *Figure 2—figure supplement 2* and *Figure 4—figure supplement 3*), with an e-value threshold of $10^{-4}$. In some cases, additional homologous gene pairs were identified via shared annotation, consistent operon structure, and conserved genome organization.

## Phylogenetic tree of AcrIIA11 homologs

The gene tree in *Figure 4* and *Figure 4—figure supplement 4* was built using the homologs in NCBI (see above section) and additionally a set of homologs identified via a BLASTP search of IMG/VR (the January 1, 2018 release), a curated database of cultured and uncultured DNA viruses (*Paez-Espino et al., 2017*). An e-value cutoff of $1 \times 10^{-10}$ was used in the IMG/VR homolog search. A preliminary phylogeny of all AcrIIA11 homologs was used to select genes that optimally sample AcrIIA11 diversity for gene synthesis and anti-Cas9 activity screening. The final phylogeny contains unique NCBI homologs and the viral homologs which were selected for gene synthesis (see *Supplementary file 1* table S8 for sequences and accession numbers). Alignments were performed using the Geneious (v8) alignment tool and a maximum-likelihood tree was generated with PhyML using the LG substitution model and 100 bootstraps.

## Protein expression and purification

Codon-optimized AcrIIA4 and AcrIIA11 were cloned into pET15b to contain thrombin-cleavable 6XHis N-terminal tags, with and without C-terminal 2xStrep2 tags. All plasmids were transformed into *E. coli* BL21(DE3) RIL cells except the C-terminally-tagged AcrIIA11 variant, which was transformed into a BL21(DE3) pLysS strain. Overnight cultures were grown in LB/Ampicillin (100 µg/mL), diluted 100-fold into 1 L pre-warmed LB/Amp media, grown until an OD600 of 0.6–0.8, then incubated on ice for 30 min. IPTG was added to 0.2 mM, and the cultures where shaken for 18–20 hr at 18˚C. The cells were pelleted and stored at −20˚C. Cell pellets were resuspended in lysis/wash buffer (500 mM NaCl, 25 mM Tris, pH 7.5, 20 mM Imidazole), lysed by sonication, and centrifuged for 25 min in an SS34 rotor at 18,000 rpm for 30 min. The soluble fraction was filtered through a 5 µm filter

and incubated in batch with Ni-NTA resin (Invitrogen, Cat# R90115) at 4°C for 1 hr. The resin was transferred to a gravity filtration column and washed with at least 50 volumes of wash buffer, followed by elution in 200 mM NaCl, 25 mM Tris, pH 7.5, 200 mM Imidazole. The buffer was exchanged into 200 mM NaCl/25 mM Tris, pH 7.5 by concentrating and diluting using an Amicon filter (EMD Millipore, 10,000 MWCO). Biotinylated thrombin (EMD Millipore) was added (1 U per mg of protein) and incubated for 16 hr at 18°C. Streptavidin-agarose was then used to remove thrombin according to the manufacturer's instructions (EMD Millipore). The proteins were diluted to 150 mM NaCl, 25 mM Tris pH 7.5, loaded onto a 1 mL HiTrapQ column (GE Life Sciences), and eluted by a sodium chloride gradient (150 mM to 1 M over 20 mL). Peak fractions were pooled and concentrated, bound to Ni-NTA to remove any uncleaved proteins, and the flow-through was purified via size exclusion chromatography (using Superdex75 16/60 (GE HealthCare) or SEC650 (BioRad) columns) in 200 mM NaCl, 25 mM Tris, pH 7.5, 5% glycerol. *Figure 5A* depicts size exclusion chromatography data for thrombin-cleaved AcrIIA11 lacking a 2xStrep2 tag. Peak fractions were pooled, concentrated, flash frozen as single-use aliquots in liquid nitrogen, and stored at −80°C.

Plasmid pMJ806 (addgene #39312) was used to express SpyCas9 containing an N-terminal 6XHis-MBP tag followed by a TEV protease cleavage site. The protein was expressed and purified over Ni-NTA as described above. The eluate from the Ni-NTA resin was dialyzed into 25 mM Tris, pH 7.5, with 300 mM NaCl, 1 mM DTT, and 5% glycerol. It was simultaneously cleaved overnight with homemade TEV protease at 4°C. The cleaved protein was loaded onto a 1 mL heparin HiTrap column (GE) in 300 mM NaCl, 25 mM Tris, 7.5 and eluted in a gradient extending to 1 M NaCl/25 mM Tris, 7.5. Pooled fractions were concentrated and buffer-exchanged with 200 mM NaCl, 25 mM Tris (pH 7.5), and 20 mM imidazole, and bound to Ni-NTA to remove uncleaved fusion proteins. The flow-through was purified over a Superdex 200 16/60 column (GE Healthcare) equilibrated in buffer containing 200 mM NaCl, 25 mM Tris (pH 7.5), 5% glycerol, and 2 mM DTT. Peak fractions were pooled, concentrated to 3–6 mg/mL, flash frozen as single-use aliquots in liquid nitrogen, and stored at −80°C. A second SpyCas9 variant was also expressed and purified from a pET28a backbone (addgene #53261) as described above except that tags were not removed. This second SpyCas9 variant includes an N-terminal 6xHis tag, a C-terminal HA-tag, and a C-terminal NLS sequence. DNA cleavage assays and Acr pulldowns were performed using tagged SpyCas9, whereas EMSAs used the untagged version. EMSAs were also performed using the tagged SpyCas9 variant and the results did not differ between SpyCas9 purifications.

## sgRNA Generation

sgRNA was generated by T7 RNA polymerase using Megashortscript Kit (Thermo Fisher #AM1354). Double-stranded DNA template was generated by a single round of thermal cycling (98°C for 90 s, 55°C for 15 s, 72°C for 60 s) in 50 µl reactions using Phusion PCR polymerase mix (NEB) containing 25 pmol each of the following ultramers (the protospacer-matching sequence is underlined):

GAAATTAATACGACTCACTATAGGTAATGAAATAAGATCACTACGTTTTAGAGCTAGAAATAG-CAAGTTAAAATAAGGCTAGTCCG and AAAAAAGCACCGACTCGGTGCCACTTTTTCAAGTTGA TAACGGACTAGCCTTATTTTAACTTGC.

The dsDNA templates were purified using an Oligo Clean and Concentrator Kit (ZymoResearch) and quantified by Nanodrop. Transcription reactions were digested with DNAse, extracted with phenol-chloroform followed by chloroform, ethanol precipitated, resuspended in RNase free water and stored at −20°C. RNA was quantified by Nanodrop and analyzed on 15% acrylamide/TBE/UREA gels.

## DNA cleavage assay

The buffer used in DNA cleavage reactions was NEB buffer 3.1 (100 mM NaCl, 50 mM Tris-HCl, pH 7.9, 10 mM MgCl2, 100 µg/mL BSA); proteins were diluted in 130 mM NaCl, 25 mM Tris, pH 7.4, 2.7 mM KCl. SpyCas9 (0.4 µM) and AcrIIA11 (0.4–12.8 µM) were incubated for 10 min at room temperature before the reaction was started by simultaneously adding 0.4 µM sgRNA and 4 nM linearized plasmid (2.6 Kb) and transferring reactions to a 37°C water bath. After 10 min at 37°C, the reaction was stopped by adding 0.1% SDS and 50 mM EDTA. Reactions were then run on a 1.25% agarose gel containing ethidium bromide at 115V for 2 hr at room temperature. Gels were imaged using the ethidium bromide detection protocol on a BioRad Chemidoc gel imager.

## Pull-down assays using Strep-tagged AcrIIA4 and AcrIIA11

The binding buffer for pull-down assays was 200 mM NaCl, 25 mM Tris (pH 7.5); protein dilutions were made in the same buffer. In 20 µl binding reactions, 160 pmol of SpyCas9 and sgRNA were incubated for 20 min at room temperature, followed by incubation with 210 pmol of strep-tagged Acr for an additional 20 min at room temperature. 50 µl of a 10% slurry of Streptactin Resin (IBA biosciences #2-1201-002) equilibrated in binding buffer was added to the binding reactions and incubated at 4°C on a nutator. Thereafter all incubations and washes were carried out at 4°C or on ice. The beads were washed a total of four times, including one tube transfer, by centrifuging 1 min at 2000 rpm, carefully aspirating the supernatant with a 25 gauge needle and resuspending the beads in 100 µl binding buffer. After the final bead aspiration, Strep-tagged proteins were eluted by resuspending in 40 µl of 1X BXT buffer (100 mM Tris-Cl, 150 mM NaCl, 1 mM EDTA, 50 mM Biotin, pH 8.0) and incubated for 15 min at room temperature. The beads were spun and 30 µl of the supernatant was carefully removed and mixed with 2X SDS Sample Buffer (Novex). Proteins were then separated by SDS PAGE on BOLT 4–12% gels in MES buffer (Invitrogen), followed by Coomassie staining.

## Electrophoretic mobility shift assays

Reactions were carried out in EMSA binding buffer (56 mM NaCl, 10 mM Tris, pH 7.4, 1.2 mM KCl, 5% glycerol, 1 mM DTT, 2 mM EDTA, 50 µg/ml heparin, 100 µg/ml BSA, 0.01% Tween-20); proteins were diluted in 130 mM NaCl, 25 mM Tris, pH 7.4, 2.7 mM KCl. Omitting MgCl2 ensured that SpyCas9 did not cleave target DNA, as previously described (*Lee et al., 2018*). DNA gel shifts used a 6FAM-labeled 60-mer or 36-mer target dsDNA (see *Supplementary file 1* table S7 for sequence) and was visualized on a BioRad Chemidoc gel imager. All incubations were carried out at room temperature. SpyCas9 and sgRNA (each at 2 µM or, in the case of *Figure 6—figure supplement 1A*, 2.25 µM) were incubated for 25 min, followed by addition of Acrs (2-16x molar excess over SpyCas9) for 20 min, followed by addition of 20 nM dsDNA template and incubation for 20 min. Samples were loaded on an 8% acrylamide/0.5X TBE gel that was pre-run (30 min, 90 V, 4°C). Reactions were resolved for 160 or 120 min (*Figure 6* and *Figure 6—figure supplement 1A*, respectively) at 4°C, 90 V in 0.5X TBE buffer. For sgRNA EMSA experiments (*Figure 5—figure supplement 1* and *Figure 6—figure supplement 1C and D*), AcrIIA11 (at 1–32 µM) was incubated with 2 µM SpyCas9 for 15 min, followed by incubation with 0.2 µM sgRNA for 20 min. The sgRNA was melted at 95°C for five minutes and then slowly cooled at 0.1 °C/s to promote proper folding prior to use. sgRNAs used in EMSAs were verified for function in SpyCas9 cleavage assays. The same buffers were used as in DNA EMSAs, except that 3 mM MgCl2 was included. Samples were run for 225 min under the gel conditions described above. The gels were post-stained with a 1:10,000 dilution of SYBR-Gold (Invitrogen) in 0.5X TBE to visualize RNA. Native gels with only apo-SpyCas9 and/or AcrIIA11 (*Figure 6—figure supplement 1C and D*) were prepared and run exactly as described for the sgRNA EMSAs, with nuclease-free water replacing sgRNA. They were then Coomassie-stained or analyzed by SpyCas9 Western blot.

## Western blots

To determine the extent to which SpyCas9 migrated through native gels, we transferred total protein to a 0.2 µM nitrocellulose membrane using the Bio-Rad Trans-Blot Turbo system (25 V, 1.3 A for 10 min). Membranes were washed with wash buffer (PBS/0.1% Triton-X) before incubation with a 1:5000 dilution of primary antibody (monocolonal, N-terminal anti-SpyCas9, Diagenode cat #C15200229-50) in Licor Odyssey Blocking Solution (part no. 927–40000). Membranes were left shaking for either two hours at room temperature or overnight at 4°C. Then, the membrane was washed four times (ten-minute washes) with wash buffer before a 30 min, room-temperature incubation with a secondary antibody conjugated to an infrared dye (IR800 donkey, anti-mouse IgG, Licor cat# 926–32212). Following three additional washes, blots were imaged on a Licor Odyssey CLx.

To verify Acr expression in 293 T cells, we performed Western blots as follows on cells collected 72 hr post transfection. Samples were transfected as described in 'Mammalian Genome Editing by SpyCas9' – the *CACNA1D* or non-targeting sgRNAs were used to load SpyCas9. Equal cell numbers were harvested across samples and proteins extracted using RIPA buffer with protease inhibitors. Sample input was normalized using a Bradford protein assay that was calibrated with BSA standards.

For each sample, 20 ug of total protein was run on a 4–20% Mini-PROTEAN TGX Precast Protein Gel (Bio-Rad). Proteins were then transferred onto a PVDF membrane using a wet Mini Trans-Blot Cell per the manufacturer's instructions (Bio-Rad). Membranes were blocked in LICOR Odyssey Blocking Buffer (Neta Scientific) incubated with the following primary antibodies overnight: rabbit anti-HA antibody (1:5000; ICL Lab) for anti-CRISPRs, mouse anti-FLAG M2 antibody (1:2000; Sigma-Aldrich) for SpyCas9, and mouse anti-beta tubulin antibody (1:2000; Thermo Fisher) as a loading control. The membrane was washed three times with PBST (1X PBS, 0.1% Tween 20) for 10 min and then incubated with the following secondary antibodies for 2 hr: goat anti-mouse IRDye680 conjugated antibody (1:10,000) and goat anti-rabbit IRDye800 conjugated antibody (1:10,000). The membrane was washed again with PBST three times for 10 min and imaged on the Odyssey Li-Cor.

## Mammalian genome editing by SpyCas9

All oligonucleotides and human codon-optimized gene fragments were purchased from IDT and are listed in *Supplementary file 1* table S7. The EF1a promoter-driven SpyCas9 expression vector was purchased from Addgene [Plasmid #98293; (*Stringer et al., 2019*). The CMV promoter-driven AcrIIA4 expression vector was also purchased from Addgene [Plasmid #113038; (*Bubeck et al., 2018*). Previously-published sgRNAs for *CACNA1D* (GCAGGAGUAUUUCAGUAGUG) and *EMX1* (GAGUCCGAGCAGAAGAAGAA) were incorporated into the SpyCas9 expression vector using Golden Gate cloning via BsmBI cut sites (*Wang et al., 2018*). The non-target sgRNA (GUAUUAC UGAUAUUGGUGGG) was cloned identically. The AcrIIA4 expression vector was modified to express AcrIIA11 variants with N-terminal NLS and HA tags using the HiFi Assembly Kit (NEB), via suggested protocols.

HEK293T cells were maintained in DMEM (Thermo Fisher/Gibco) containing phenol red, 4 mM L-glutamine, 110 mg/L sodium pyruvate, 4.5 g/L D-glucose, and supplemented with 10% (v/v) FBS (Thermo Fisher/Gibco) and 100 U/mL penicillin + 100 µg/mL streptomycin (Thermo Fisher/Gibco). Cell lines were authenticated and tested for mycoplasma contamination before use via the Mycoplasma Detection Kit (Southern Biotech). Transient transfections were performed with Lipofectamine 2000 (Life Technologies), according to the manufacturer's instructions. Approximately 350,000 cells were seeded in each well of a 12-well plate 24 hr prior to transfection to allow cells to become 60–70% confluent at the time of transfection. Wells were transfected with either the anti-CRISPR expression vector, the SpyCas9/sgRNA expression vector, or both using 500 ng for each vector (3:1 Acr: SpyCas9/sgRNA plasmid ratio) and either 1.5 µL or 3 µL of Lipofectamine (3 µL per µg of DNA).

Cells were collected and pelleted 72 hr post transfection for genomic DNA extraction using the Wizard Genomic DNA Purification Kit (Promega). The target locus was PCR-amplified using Accu-Prime Pfx high-fidelity DNA polymerase (Thermo Fisher) and the following PCR conditions: 95°C for 2 min, 35 cycles of 98°C for 15 s + 64°C for 30 s + 68°C for 2 min, and 68°C for 2 min. Reaction-specific primers and conditions are listed in *Supplementary file 1* table S7. Indel frequencies at the Spy-Cas9 target site were assessed via a T7E1 assay with the EnGen Mutation Detection Kit (NEB), using manufacturer's recommendations. Reaction products were analyzed on a 1.5% SeaKem GTG agarose gel (Lonza) and imaged with the InGenuis3 (Syngene). For calculating indel percentages from gel images, bands from each lane were quantified with GelAnalyzer (version 2010a freeware). Peak areas were measured and percentages of insertions and deletions [Indel(%)] were calculated using the formula: *Indel(%)=100 × (1 – (1 – Fraction cleaved)\*0.5), where Fraction cleaved = (Σ (Cleavage product bands))/(Σ (Cleavage product bands + PCR input band)).*

## Acknowledgements

We thank Cara Forsberg, Tera Levin, Courtney Schroeder, Gerry Smith, Jeannette Tenthorey, and Janet Young for comments on the manuscript, Gautam Dantas for the functional metagenomic libraries, Andrew Taylor for purified RecBCD, and the Fred Hutchinson Cancer Research Center Genomics core facility for Illumina sequencing. This work was supported by a postdoctoral fellowship awarded to KJF by the Helen Hay Whitney Foundation, and by a Seattle University summer faculty fellowship to BKK. Further support for this work includes discretionary funding from the Fred Hutchinson Cancer Research Center to BLS and awards from the National Institutes of Health (R01GM105691 to BLS, R01GM124131 to IJF, and F31GM125201 to KED), the Welch Foundation (F-1808 to IJF), and the Howard Hughes Medical Institute to HSM. The funders played no role in study

design, data collection and interpretation, or the decision to publish this study. HSM is an Investigator of the Howard Hughes Medical Institute.

# Additional information

## Funding

| Funder | Grant reference number | Author |
|---|---|---|
| Helen Hay Whitney Foundation | | Kevin J Forsberg |
| Seattle University | | Brett K Kaiser |
| National Institute of General Medical Sciences | R01 GM105691 | Barry L Stoddard |
| Fred Hutchinson Cancer Research Center | | Barry L Stoddard |
| Howard Hughes Medical Institute | | Harmit S Malik |
| National Institute of General Medical Sciences | F31 GM125201 | Kaylee E Dillard |
| National Institute of General Medical Sciences | R01 GM124131 | Ilya J Finkelstein |
| Welch Foundation | F-1808 | Ilya J Finkelstein |

The funders had no role in study design, data collection and interpretation, or the decision to submit the work for publication.

## Author contributions

Kevin J Forsberg, Conceptualization, Data curation, Formal analysis, Funding acquisition, Validation, Investigation, Visualization, Methodology, Writing—original draft, Project administration, Writing—review and editing; Ishan V Bhatt, Kamyab Javanmardi, Kaylee E Dillard, Investigation, Writing—review and editing; Danica T Schmidtke, Validation, Investigation, Writing—review and editing; Barry L Stoddard, Harmit S Malik, Conceptualization, Resources, Supervision, Funding acquisition, Project administration, Writing—review and editing; Ilya J Finkelstein, Resources, Supervision, Funding acquisition, Project administration; Brett K Kaiser, Conceptualization, Resources, Funding acquisition, Validation, Investigation, Methodology, Project administration, Writing—review and editing

## Author ORCIDs

Kevin J Forsberg (iD) https://orcid.org/0000-0002-1545-8925
Ilya J Finkelstein (iD) http://orcid.org/0000-0002-9371-2431
Harmit S Malik (iD) http://orcid.org/0000-0001-6005-0016

## Decision letter and Author response

Decision letter https://doi.org/10.7554/eLife.46540.029
Author response https://doi.org/10.7554/eLife.46540.030

# Additional files

## Supplementary files

• Supplementary file 1. Supplementary tables S1 to S8.
DOI: https://doi.org/10.7554/eLife.46540.024

• Transparent reporting form
DOI: https://doi.org/10.7554/eLife.46540.025

## Data availability

Sequencing data is available on NCBI BioProject under accession number PRJNA526924.

The following dataset was generated:

| Author(s) | Year | Dataset title | Dataset URL | Database and Identifier |
|---|---|---|---|---|
| Forsberg KJ, Bhatt IV, Schmidtke DT, Javanmardi K, Dillard KE, Stoddard BL, Finkelstein IJ, Kaiser BK, Malik HS | 2019 | Functional Metagenomic Selection Reveals Potent Cas9 Inhibitors in Human Oral and Fecal Metagenomes | https://www.ncbi.nlm.nih.gov/bioproject/PRJNA526924 | NCBI BioProject, PRJNA526924 |

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
