## [Decision Letter]

Thank you for submitting your article "Functional metagenomics-guided discovery of potent Cas9 inhibitors in the human microbiome" for consideration by *eLife*. Your article has been reviewed by three peer reviewers reviewers, including Peter Turnbaugh as the Reviewing Editor and Reviewer #1, and the evaluation has been overseen by Wendy Garrett as the Senior Editor. The following individual involved in review of your submission has also agreed to reveal their identity: Joe Bondy-Denomy (Reviewer #2).

The reviewers have discussed the reviews with one another and the Reviewing Editor has drafted this decision to help you prepare a revised submission.

Summary:

Forsberg and colleagues present a functional metagenomic selection to enrich for DNA fragments that encode for anti-CRISPR (acr) proteins. The application of this method to oral and gut metagenomic libraries led to the discovery and functional characterization of a novel type of acr that acts in a manner distinct from published acrs and is found in multiple genomes from the dominant taxonomic groups present in the human gut. This platform is incredibly promising for accelerating the unbiased identification of acrs and providing new insights into the interactions between viruses and bacteria within the human body.

Essential revisions:

This paper is clearly written and describes an approach that should be widely applicable with a new finding. However, the novelty is reduced somewhat by the following factors: (1) Uribe et al. report a similar screen with more proteins discovered, (2) reporting only a single Acr does not seem to fully harness the elegance and throughput of the screen that the authors established and optimized, (3) it is expected that new anti-CRISPRs with novel sequences are out there, (4) while the mechanism of action appears to be novel to type II-A (as described by the authors), two concerns remain: a) the in vitro data generate confusion about how it is actually working, and the mechanism is not novel per se, as the authors clearly state in the discussion, where cleavage inhibition has been shown with AcrIIC1 (and missing AcrF3/AcrE1, conceptually similar).

The manuscript could be greatly improved by doing one or more of the following:

1) Identify more genes that inhibit SpCas9 with the existing screen, or by looking at the neighbors of AcrIIA11 shown in Figure 2—figure supplement 2.

2) Provide further mechanistic details, which could be obtained by exploring the non-specific DNA interactions, shortening the DNA substrate, etc.

3) Test the spectrum of activity of this new protein with assays against orthologous Cas9 proteins.

4) Perform in vivo (bacterial or eukaryotic) experiments that test the predictions of the mechanistic work (i.e. confirming that this Acr would turn SpCas9 into dSpCas9 with phage or reporter experiments)

---

## [Author Response]

Essential revisions:This paper is clearly written and describes an approach that should be widely applicable with a new finding. However, the novelty is reduced somewhat by the following factors: (1) Uribe et al. report a similar screen with more proteins discovered, (2) reporting only a single Acr does not seem to fully harness the elegance and throughput of the screen that the authors established and optimized, (3) it is expected that new anti-CRISPRs with novel sequences are out there, (4) while the mechanism of action appears to be novel to type II-A (as described by the authors), two concerns remain: a) the in vitro data generate confusion about how it is actually working, and the mechanism is not novel per se, as the authors clearly state in the discussion, where cleavage inhibition has been shown with AcrIIC1 (and missing AcrF3/AcrE1, conceptually similar).

The reviewers are correct in that Uribe et al. also employ a functional metagenomics approach to discover Acrs. Despite the thematic overlap, the mechanics or our screen differ significantly from theirs. Moreover, we feel that our work is significantly distinguished from Uribe et al. by the depth of analysis we provide for a single Acr family, AcrIIA11, which is distinct from any previous discovery (including those reported in Uribe et al). In addition to the extensive discussion of Uribe et al. in our original submission, we now highlight their work in our introduction to make the thematic overlap more transparent.

We have updated our Discussion section to avoid implying that Acrs with novel sequence are a surprise finding. Instead, we emphasize in our introduction and discussion that many new Acrs likely exist in nature and that functional metagenomics is a promising strategy for their discovery.

We address the remainder of the comments below.

The manuscript could be greatly improved by doing one or more of the following:1) Identify more genes that inhibit SpCas9 with the existing screen, or by looking at the neighbors of AcrIIA11 shown in Figure 2—figure supplement 2.

To address this concern, as suggested by one of the reviewers (below), we adopted a ‘guilt-by-association’ approach first highlighted in Pawluk et al., (2016). We cloned 17 genes that flank AcrIIA11a.1 and AcrIIA11a.2 in host genomes and tested them for SpyCas9 antagonism using a plasmid protection assay (Figure 4—figure supplement 3B,3C). We chose these loci because they also contained putative homologs of the anti-CRISPRassociated (aca) gene aca4. Unfortunately, although a few of these 17 genes protected a target plasmid marginally better than GFP, none were as potent as bona fide SpyCas9 inhibitors. We report the sequence and activity of these candidate genes in our revised manuscript. We acknowledge that these partial inhibitors may exhibit anti-CRISPR activity in other situations, (e.g., in different hosts or against different CRISPR-Cas systems), consistent with their genomic context, and discuss this possibility.

2) Provide further mechanistic details, which could be obtained by exploring the non-specific DNA interactions, shortening the DNA substrate, etc.

We have performed several additional experiments to respond to this request:

First, we explored these non-specific DNA interactions (Figure 6—figure supplement 1B) and have performed EMSAs with a short DNA substrate (Figure 6—figure supplement 1A). These data reveal two key properties of AcrIIA11:

i) AcrIIA11 binds the SpyCas9/sgRNA/dsDNA ternary complex to create the ‘super-shift’ in Figure 6, instead of binding adjacent sDNA.

ii) AcrIIA11 is stimulated to bind dsDNA in the presence of SpyCas9.

Second, in response reviewer point #25 (below), we ruled out sgRNA binding as a potential mechanistic explanation for AcrIIA11’s SpyCas9 inhibition (Figure 5—figure supplement 1).

Third, we demonstrate that AcrIIA11 triggers a similar ‘super-shift’ using apo-SpyCas9, sgRNA-loaded SpyCas9, and a SpyCas9/sgRNA/dsDNA ternary complex (Figure 6—figure supplement 1). These findings suggest that AcrIIA11 binds a SpyCas9 motif preserved across all three conformations.

While we do not yet have a comprehensive understanding of AcrIIA11’s mode of SpyCas9 inhibition, our new data give us a clearer picture of AcrIIA11’s activity, moving us closer to that goal. These data also support our central conclusion about AcrIIA11’s mechanism – that it is different than any other type II-A Acr. Taken together, we believe that our additional biochemical experiments add enough new mechanistic information to satisfactorily address this point.

3) Test the spectrum of activity of this new protein with assays against orthologous Cas9 proteins.

We thank the reviewers for this suggestion. In response to this request, we performed an entirely new series of experiments testing AcrIIA11 homologs against the type II-A CRISPR-Cas9 system from Trepenoma denticola in a plasmid protection assay. We find that many of our A11 homologs protect against TdeCas9, described in Figure 4—figure supplement 4. These data support our hypothesis that AcrIIA11 acts broadly against Cas9.

*4) Perform* in vivo *(bacterial or eukaryotic) experiments that test the predictions of the mechanistic work (i.e. confirming that this Acr would turn SpCas9 into dSpCas9 with phage or reporter experiments).*

Although we like the hypothesis that AcrIIA11 might turn SpyCas9 into dCas9, our mechanistic findings also leave open other possibilities. Nevertheless, to test the effects of AcrIIA11 inhibition in vivo, we now include an entirely new set of experiments (performed in collaboration with the Finkelstein lab) that show AcrIIA11 does not rely on bacteria-specific factors. These experiments (described in new Figure 7) show that AcrIIA11 can inhibit SpyCas9 in a human cell line, HEK-293T, but only at some loci. This suggests some feature of the target site (for instance, its chromatin state) impacts AcrIIA11’s activity. Additionally, AcrIIA11’s effectiveness in human cells widens its potential uses for modifying new Cas9-based tools in medicine and research.